# A Survey on Industrial Anomaly Synthesis

## Abstract

This paper presents a comprehensive review of industrial anomaly synthesis (IAS). Existing surveys on industrial anomalies mainly focus on anomaly detection, while anomaly synthesis is typically treated as an auxiliary component rather than as an independent topic. However, owing to its increasing importance in data augmentation, downstream model training, and controllable industrial inspection, IAS has become a research direction of growing interest. To address the lack of a dedicated review, we survey a broad range of representative methods and organize them into four paradigms: hand-crafted synthesis, distribution hypothesis-based synthesis, generative model (GM)-based synthesis, and vision-language model (VLM)-based synthesis. We further establish a dedicated taxonomy for IAS, which supports more systematic comparison across methods and offers a clearer view of the field's development. Beyond methodological categorization, we summarize the datasets, benchmarks, and evaluation metrics commonly adopted in IAS, and review recent advances in multimodal anomaly synthesis that remain underexplored in prior surveys. We also provide deployment-oriented comparisons and practical guidance by analyzing input requirements, output forms, controllability, cost, downstream tasks, and practical limitations across IAS subcategories. Overall, this survey provides a structured understanding of existing IAS methods, evaluation settings, practical trade-offs, current limitations, and promising future directions, and is intended to serve as a reference for subsequent research in this area. More resources are available at https://anonymous.4open.science/status/IAS.

## 1 Introduction

Image anomaly detection plays an important role in manufacturing because it helps identify abnormal products and thereby supports product quality, production safety, and process reliability. In practical industrial settings, however, building effective image anomaly detection systems usually relies on a sufficient number of high-quality annotated abnormal samples for training, and obtaining such samples is often costly and difficult. The main reasons are summarized as follows: (1) The proportion of abnormal products is usually very low in large-scale manufacturing, resulting in a natural scarcity of real abnormal samples. (2) Many industrial anomaly patterns, such as microscopic cracks, fine scratches, concealed contaminants, or internal structural anomalies, require specialized inspection equipment, *e.g.*, high-magnification microscopes, X-ray systems, or infrared devices, which significantly increases the cost of data acquisition. (3) High-quality annotation further requires domain expertise and careful analysis. In many cases, skilled professionals are needed to identify abnormal regions accurately, and some samples may even require multimodal annotation, which further increases the time and labor cost.

To alleviate the shortage of real abnormal samples, a growing number of industrial anomaly synthesis (IAS) methods have been developed for data augmentation and downstream model training. As shown in Fig. 1, IAS has attracted rapidly increasing attention in recent years, reflecting its growing role in industrial inspection. This trend is closely related to practical demand. In many real applications, it is often not sufficient to only determine whether a sample is abnormal. Instead, practitioners increasingly expect IAS to support diverse anomaly synthesis, segmentation-oriented supervision, and more controllable training data construction under specific industrial contexts. IAS is also expected to facilitate benchmark construction, stress testing, and failure-case analysis for downstream inspection systems. In this sense, IAS is gradually evolving

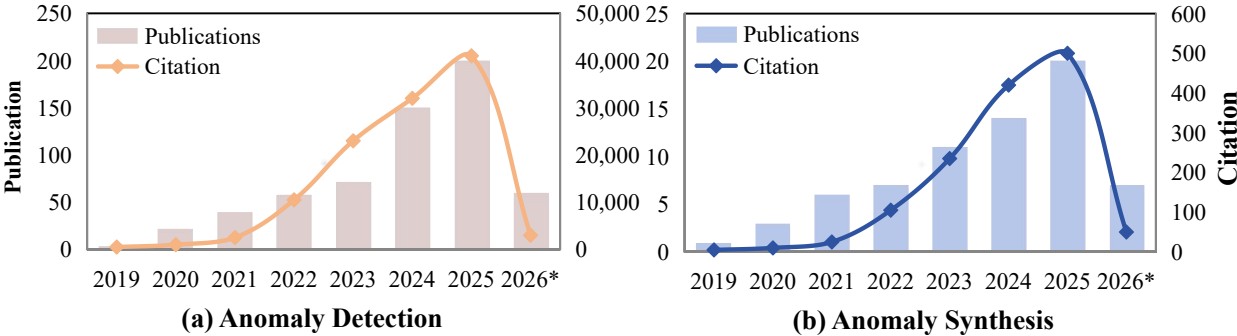

Figure 1: Publication and citation trends of anomaly detection and anomaly synthesis from 2019 to 2026* (2026* denotes partial-year statistics up to March 30, 2026). Anomaly detection shows steady growth, while anomaly synthesis has attracted increasing attention.

from a simple augmentation strategy into a more task-oriented component for industrial model development and evaluation. Nevertheless, despite this progress, current IAS methods still cannot fully satisfy practical industrial requirements. Their main limitations can be summarized into three aspects:

(1) **Limited Coverage of Abnormal Distribution.** In practical industrial scenarios, the number of available abnormal samples for a specific anomaly type is often very small. As a result, existing methods usually capture only a limited part of the underlying anomaly distribution, which restricts diversity and weakens generalization. This issue is particularly evident for long-tail or highly irregular anomalies, such as subtle cracks, local corrosion, stains, punctures, or composite anomalies that vary greatly in shape, scale, and texture.

(2) **Difficulty in Synthesizing Realistic Abnormal Samples.** Real industrial abnormal patterns are often structurally complex and visually diverse, while also being tightly coupled with local material properties, imaging conditions, and background context. For example, a fine scratch on a polished metal surface, a contamination spot on transparent packaging, or internal anomalies revealed by X-ray imaging may each exhibit very different visual characteristics. This makes it difficult for synthesis methods to preserve both realism and consistency. As a result, synthesized results may still contain unrealistic textures, missing details, or unnatural transition boundaries.

(3) **Limited Use of Multimodal Information for Controllable Synthesis.** In industrial scenarios, useful cues may come from multiple modalities, such as text descriptions, masks, spatial priors, reference images, or other auxiliary signals. These cues are important when users need to specify where an anomaly should appear, what type of anomaly should be synthesized, or how strong the abnormal pattern should be. However, how to effectively incorporate such multimodal information into IAS remains insufficiently explored, especially for controllable and realistic anomaly synthesis.

Although IAS has developed rapidly, a dedicated and systematic review remains lacking. Existing anomaly-related surveys are not completely unrelated to anomaly synthesis; many of them mention data augmentation, generative modeling, or synthetic anomalies within broader industrial anomaly detection pipelines. However, these discussions are usually embedded in detection-oriented narratives, where anomaly synthesis is treated as an auxiliary tool rather than as an independent methodological problem. As summarized in Table 1, prior surveys have provided valuable discussions on visual inspection, data augmentation, generative modeling, and industrial anomaly detection, but they do not yet offer a unified synthesis-centered understanding of IAS itself. In particular, current reviews usually cover only part of the synthesis literature, rarely organize methods according to their dominant anomaly-forming mechanisms, and provide limited discussion of the recent shift toward multimodal and VLM-based controllable synthesis. Consequently, it remains difficult to systematically compare how different IAS methods construct anomalies, what inputs and supervision signals they require, and how their outputs support downstream tasks. It also remains difficult to clearly trace how the field has evolved from early rule-based perturbation and latent-space deviation to recent image-space generation, local editing, and multimodal controllable synthesis.

Table 1: Comparison of previous surveys and our survey.

| Ref. | Year | Perspective | IAS Cat. | Multimodal | Description |
|---|---|---|---|---|---|
| (Chen et al., 2021) | 2021 | Detection | 0 | ✗ | Visual inspection review with emphasis on augmentation and practical pipelines. |
| (Xia et al., 2022) | 2022 | Detection | 4 | ✗ | GAN-centric review covering GAN-based techniques related to AD and synthesis. |
| (Liu et al., 2024) | 2024 | Detection | 2 | ✗ | Broad anomaly detection survey; anomaly synthesis is not a primary focus. |
| (Cao et al., 2024) | 2024 | Detection | 4 | ✗ | Discusses anomaly-related topics and mentions multimodal cues; IAS is not systematically reviewed. |
| (Mao et al., 2025) | 2025 | Detection | 1 | ✗ | Industrial anomaly detection survey covering paradigms, benchmarks, and metrics; includes synthesis-related discussion. |
| (Li et al., 2025) | 2025 | Detection | 1 | ✗ | Industrial visual anomaly detection survey covering learning strategies, generative modeling, and multimodal/VLM-based detection methods. |
| **Ours** | **2026** | **Synthesis** | **10** | ✓ | **Dedicated IAS taxonomy; reviews ∼64 representative IAS methods across paradigms; summarizes benchmarks and practical trade-offs.** |

Motivated by this gap, this paper presents a dedicated survey on industrial anomaly synthesis. Rather than discussing anomaly synthesis only as a supporting tool for anomaly detection, we study IAS as an independent methodological direction with its own synthesis mechanisms, benchmark choices, evaluation settings, and practical trade-offs.

To establish a representative literature corpus for this review, we identify IAS-related studies through keyword-based search, venue-oriented screening, and citation snowballing. The search uses keywords such as "industrial anomaly synthesis", "defect generation", "anomaly generation", "synthetic anomaly", "controllable anomaly synthesis", and "industrial defect image generation". We mainly focus on works available from 2019 to March 30, 2026 across major computer vision, machine learning, artificial intelligence, multimedia, and industrial inspection venues, while retaining earlier foundational studies and benchmarks when necessary to trace the methodological development of IAS. We primarily include studies whose main technical contribution explicitly involves synthesizing abnormal images, anomaly masks, image–mask pairs, or anomaly-oriented latent/feature representations for industrial downstream tasks such as anomaly detection, localization, inspection, or segmentation. We exclude general anomaly detection papers that only use standard data augmentation without a dedicated anomaly synthesis mechanism, works outside industrial visual inspection, and duplicate preprint versions when a peer-reviewed version is available. To improve coverage and representativeness,

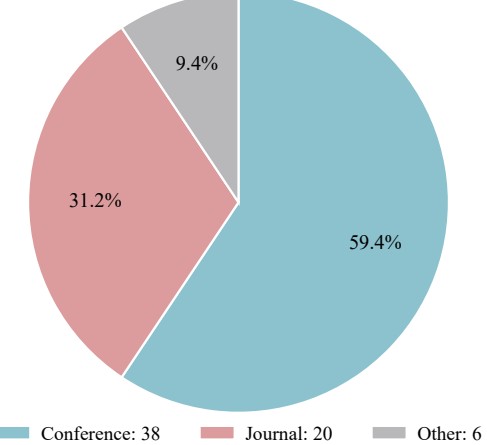

Figure 2: Publication-source distribution of the 64 representative IAS methods reviewed in this survey. Conference papers constitute 59.4%, primarily from major venues such as CVPR, AAAI, ECCV, and ICCV. Journal articles account for 31.2% and span established venues including IEEE TIP, IEEE TII, IEEE TIM, IEEE TASE, and Pattern Recognition. The remaining 9.4% come from other sources, mainly preprints.

we also consider influential studies, methods with publicly available implementations, and recent arXiv or OpenReview preprints that introduce important IAS methodologies, benchmarks, or evaluation protocols. The resulting corpus covers 64 representative IAS methods from major conference proceedings, peer-reviewed

journals, and a small number of other sources, as summarized in Fig. 2. Since IAS is rapidly evolving, especially with concurrent 2026 works, this survey is intended to be representative rather than exhaustive.

Based on this literature corpus, we organize existing IAS methods into four paradigms, namely hand-crafted synthesis, distribution hypothesis-based synthesis, generative model (GM)-based synthesis, and vision-language model (VLM)-based synthesis. We first present the proposed taxonomy and clarify the key methodological distinctions among these paradigms, and then summarize commonly used datasets, benchmarks, and evaluation protocols in IAS. Finally, we discuss the current limitations, open challenges, and promising future directions of IAS.

Overall, the main contributions of this paper are summarized as follows:

- We present a dedicated taxonomy for industrial anomaly synthesis (IAS), which organizes existing methods into four paradigms: hand-crafted synthesis, distribution hypothesis-based synthesis, generative model (GM)-based synthesis, and vision-language model (VLM)-based synthesis. This taxonomy provides a structured and fine-grained framework for understanding methodological differences, technical evolution, and practical design choices in IAS.

- We summarize commonly used datasets, benchmarks, and evaluation protocols, thereby clarifying how IAS methods are commonly developed and assessed in the literature. Based on the proposed taxonomy, we then provide a unified and systematic review of a broad range of representative IAS methods.

- We discuss recent advances, current limitations, open challenges, and promising future directions of IAS research, with particular attention to controllability, semantic alignment, multimodal conditions, and industrial applicability.

The remainder of this paper is structured as follows: Section 2 first introduces commonly used datasets, benchmarks, and evaluation protocols in IAS, and then presents the proposed taxonomy with its key paradigms and methodological distinctions. Sections 3 to 6 provide an in-depth analysis of the four primary categories of IAS, including Hand-crafted synthesis, Distribution hypothesis-based synthesis, GM-based synthesis, and recently emerging VLM-based synthesis. Section 7 connects the proposed taxonomy with practical deployment guidance through cross-paradigm comparisons and discusses future research directions on anomaly diversity, controllable synthesis, and multimodal integration. Section 8 concludes the survey.

## 2 Taxonomy and Benchmarking Overview

### 2.1 Taxonomy and Definitions of IAS

Fig. 3 presents the overall taxonomy of industrial anomaly synthesis (IAS). In this survey, IAS methods are organized into four main paradigms, namely hand-crafted synthesis, distribution hypothesis-based synthesis, GM-based synthesis, and VLM-based synthesis. Rather than relying on a single backbone or synthesis technique, this taxonomy distinguishes these paradigms according to their dominant synthesis mechanism, knowledge source, dependence on large-scale pre-training, task adaptation strategy, and modality. Table 2 summarizes these distinctions. Hand-crafted synthesis mainly relies on manually specified synthesis rules, while distribution hypothesis-based synthesis generates abnormal samples by perturbing normal-sample distributions. GM-based synthesis adopts generative models such as GANs, VAEs, or diffusion models, but its defining property is task-specific training, re-training, fine-tuning, or domain adaptation on industrial data. In contrast, VLM-based synthesis is built upon large-scale pre-trained vision-language models and adapts them to IAS through prompting, instruction tuning, lightweight adaptation modules, textual inversion, or few-shot references. Notably, the distinction between GM-based and VLM-based synthesis mainly lies in pre-training dependence, adaptation paradigm, and modality, rather than in the mere ability to synthesize abnormal images.

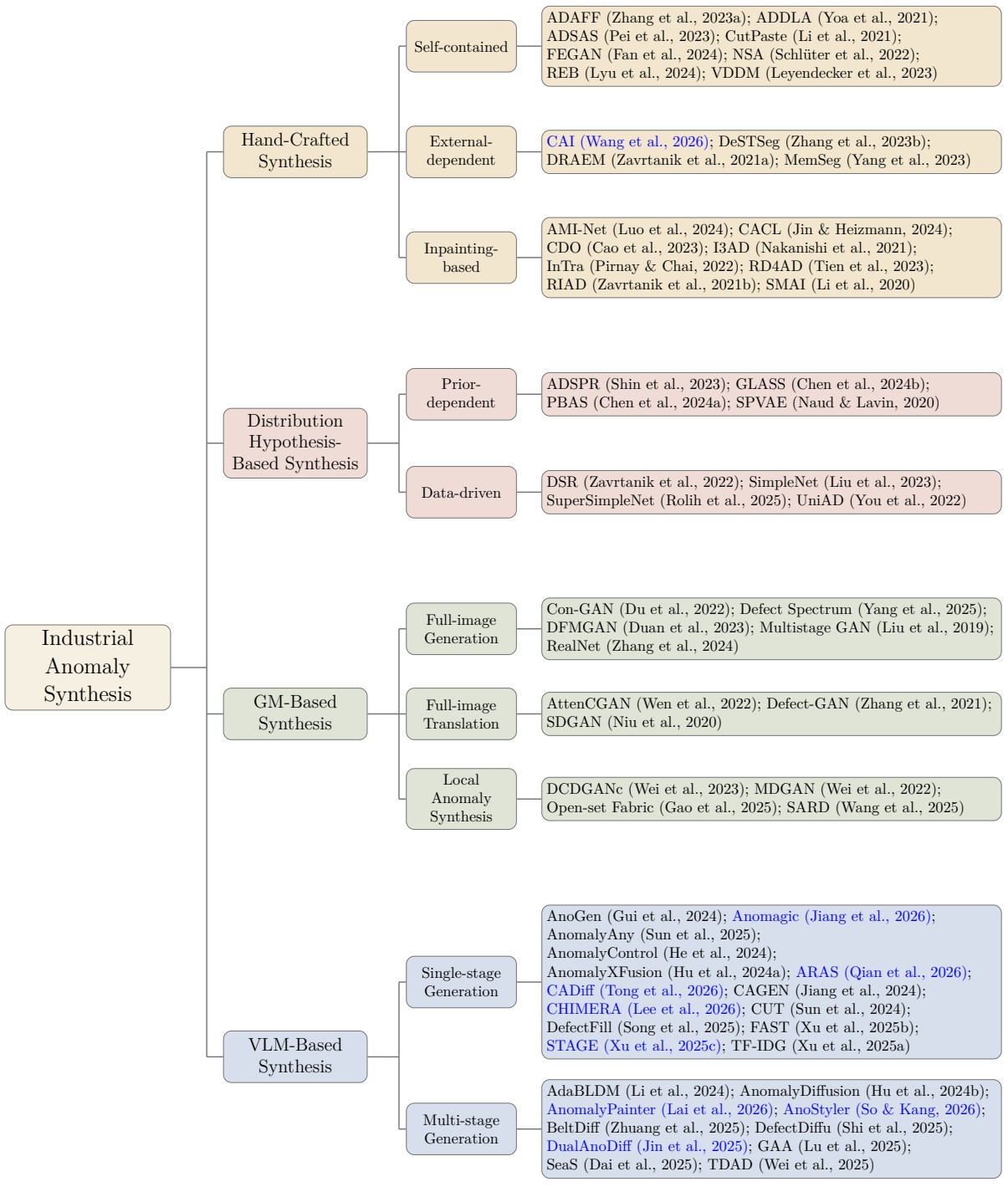

Figure 3: A taxonomy of industrial anomaly synthesis (IAS).

Hand-crafted synthesis relies on manually designed rules to construct samples with anomalies. It is usually training-free and is suitable for controlled settings where high realism and large anomaly diversity are not the primary requirements. Within this paradigm, self-contained synthesis constructs abnormal regions from available normal images through cropping, copying, rearrangement, or related operations, without introducing auxiliary external texture sources. External-dependent synthesis introduces auxiliary sources such as texture libraries, so that the synthesized content is not restricted to the image itself. Inpainting-based synthesis first masks local regions and then disrupts structural completeness by inserting noise, black patches, or

missing content. Because the abnormal content is not generated from an explicitly learned anomaly distribution, these methods are usually simple and effective for data expansion, but their realism and distributional coverage are often limited.

Distribution hypothesis-based synthesis generates anomalies by modeling normal data distributions and introducing controlled deviations, typically in latent space or feature space. Prior-dependent synthesis adopts predefined geometric assumptions, such as manifold or hypersphere structures, to characterize normality, and then synthesizes abnormal features near or beyond the normal boundary. By contrast, data-driven synthesis does not rely on explicit geometric priors. Instead, it exploits intrinsic statistical properties of the data and synthesizes anomalies through perturbation or adaptive feature manipulation, which often provides greater flexibility and diversity. As a result, this paradigm is often attractive for feature-level training augmentation and anomaly-aware representation learning, although the synthesized results may be less intuitive at the pixel level than image-space generation methods.

Table 2: Comparison of the four main paradigms for IAS.

| Criterion | Hand-crafted | Distribution hypothesis-based | GM-based | VLM-based |
|---|---|---|---|---|
| Core mechanism | Rule-guided synthesis by manually specifying texture, location, or intensity. | Distribution-guided synthesis by perturbing normal-sample distributions. | Task-specific synthesis with generative models trained or adapted on industrial data. | Multimodal semantic synthesis with large-scale pre-trained vision-language models. |
| Knowledge source | Human expertise, industrial priors, and image processing rules. | Normal-sample distributions or constrained feature representations. | Target-domain industrial data and task-specific anomaly synthesis priors learned during training. | Pre-trained vision-language knowledge and industrial data. |
| Large-scale pre-training | Not required. | Usually not required; frozen feature extractors may be used for representation. | Usually not required; synthesis ability mainly comes from target-domain training. | Required; generality and semantic knowledge come from large-scale pre-training. |
| Task adaptation | Manual adjustment of synthesis rules. | Estimation of normal patterns and construction of abnormal deviations. | Dedicated training, re-training, fine-tuning, or domain adaptation for specific categories, datasets, or anomaly types. | Prompting, instruction tuning, adapters, LoRA, textual inversion, or few-shot reference adaptation. |
| Modality | Unimodal, based on images. | Unimodal, based on images or visual representations. | Unimodal, based on images. | Multimodal, involving images, text, or other reference cues. |
| Generalization | Rules need redesign across categories or anomaly types. | Assumption-dependent; robustness decreases when normal patterns change. | Good performance within the trained domain, but limited under cross-category shifts. | Good potential for cross-category, few-shot, and semantically specified anomaly synthesis. |
| Main limitation | Limited realism and diversity; manual rules may introduce synthetic artifacts. | Weak anomaly semantics and limited image-level realism or interpretability. | Task-data dependence and possible target-domain overfitting. | High computational cost and limited real-time applicability. |

GM-based synthesis uses deep generative models, such as GANs and diffusion models, to produce more realistic abnormal samples. This paradigm can be further divided into full-image synthesis, full-image translation, and local anomaly synthesis. Full-image synthesis learns an abnormal distribution and maps random noise to abnormal samples. Full-image translation transforms normal images into abnormal ones while preserving the overall scene structure. Local anomaly synthesis modifies selected regions of normal images with generated abnormal content and emphasizes local consistency between the edited area and its surrounding background. Compared with earlier paradigms, GM-based synthesis generally offers stronger visual realism and richer anomaly appearance, but it also tends to require heavier training, more careful optimization, and higher computational cost.

VLM-based synthesis leverages large-scale pre-trained vision-language models together with multimodal conditions to produce high-quality abnormal samples. Single-stage synthesis directly synthesizes context-aware anomalies, usually through prompt engineering or lightweight fine-tuning. Multi-stage synthesis instead

adopts a staged pipeline that combines anomaly generation with additional processes such as mask generation, refinement, or sequential optimization, thereby improving realism, diversity, and downstream compatibility. The main advantage of VLM-based synthesis lies in stronger semantic controllability and better use of multimodal cues, especially when anomaly type, location, or contextual compatibility needs to be specified more explicitly. Nevertheless, applying VLM-based synthesis to IAS still faces challenges in computational efficiency, real-time applicability, and the effective incorporation of industrial-specific knowledge into open-world multimodal priors.

Although these four paradigms are presented as separate branches, they are not completely isolated in practice. Hand-crafted and distribution hypothesis-based methods usually emphasize efficiency and training utility, whereas GM-based and VLM-based methods place more emphasis on realism, diversity, and controllability. The taxonomy therefore provides a structured basis for comparing IAS methods from both methodological and practical perspectives, while avoiding a purely backbone-based framework.

## 2.2 Datasets, Benchmarks, and Evaluation Protocols

IAS results are not directly comparable unless benchmark choice, supervision setting, and evaluation context are considered together.

Table 3: Representative datasets commonly used in IAS-related studies. "Data type" indicates whether the benchmark is primarily based on real industrial imagery (Real), fully synthetic data (Syn.), or a mixture of real and synthetic components (Mixed).

| Dataset | Year | Venue | Data type (Real/Syn./Mixed) | Samples | Classes | IAS relevance H | DH | GM | VLM | Site |
|---|---|---|---|---|---|---|---|---|---|---|
| MVTec-AD (Bergmann et al., 2019) | 2019 | CVPR | Real | 5,354 | 15 | ✓ | ✓ | ✓ | ✓ | link |
| VisA (Zou et al., 2022) | 2022 | ECCV | Real | 10,821 | 12 | ✓ | ✓ | ✓ | ✓ | link |
| BTAD (Mishra et al., 2021) | 2021 | ISIE | Real | 2,830 | 3 | ✓ | ✓ | ✓ | ✗ | link |
| MPDD (Jezek et al., 2022) | 2022 | ICUMT | Real | >1,000 | 6 | ✓ | ✓ | ✓ | ✗ | link |
| KSDD (Tabernik et al., 2020) | 2020 | JIM | Real | 399 | 1 | ✓ | ✓ | ✓ | ✗ | link |
| KSDD2 (Božič et al., 2021) | 2021 | Comput. Ind. | Real | 3,335 | 1 | ✓ | ✓ | ✓ | ✗ | link |
| AITEX (Silvestre-Blanes et al., 2019) | 2019 | Autex Res. J. | Real | 245 | 7 | ✓ | ✓ | ✓ | ✗ | link |
| MVTec LOCO-AD (Bergmann et al., 2022a) | 2022 | IJCV | Real | 3,644 | 5 | ✗ | ✓ | ✓ | ✓ | link |
| PCB-Bank (Yao et al., 2024) | 2024 | ECCV | Real | 6,749 | 7 | ✗ | ✓ | ✓ | ✓ | link |
| Real-IAD (Wang et al., 2024) | 2024 | CVPR | Real | 150,000+ | 30 | ✗ | ✗ | ✓ | ✗ | link |
| MVTec 3D-AD (Bergmann et al., 2022b) | 2022 | VISAPP | Real | 4,147 | 10 | ✗ | ✗ | ✓ | ✓ | link |
| MVTec AD 2 (Heckler-Kram et al., 2025) | 2025 | arXiv | Real | 8,000+ | 8 | ✗ | ✗ | ✓ | ✓ | link |
| DAGM 2007 (Wieler et al., 2007) | 2007 | DAGM | Syn. | 16,100 | 10 | ✓ | ✗ | ✓ | ✗ | link |
| DTD-Synthetic (Aota et al., 2023) | 2023 | WACV | Syn. | >2,400 | 12 | ✓ | ✓ | ✓ | ✗ | link |
| Eyecandies (Bonfiglioli et al., 2022) | 2022 | ACCV | Syn. | 90,000 | 10 | ✗ | ✗ | ✓ | ✓ | link |
| PAD (Zhou et al., 2023) | 2023 | NeurIPS | Mixed | 11,000+ | 20 | ✗ | ✗ | ✓ | ✗ | link |
| ISP-AD (Krassnig & Gruber, 2025) | 2025 | arXiv | Mixed | 559,049 | 3 | ✗ | ✗ | ✓ | ✗ | link |

*Note:* H = Hand-crafted, DH = Distribution hypothesis-based, GM = generative model-based, VLM = vision-language model-based. IAS relevance indicates representative, non-exhaustive usage in reviewed studies.

Table 3 summarizes representative benchmarks used in IAS-related studies, covering real, synthetic, and mixed data sources. Rather than differing only in data type, these datasets also vary in scale, inspection setting, and annotation characteristics. Among real-world benchmarks, MVTec-AD (Bergmann et al., 2019) and VisA (Zou et al., 2022) are widely adopted as standard references for 2D industrial anomaly detection. Other datasets, such as BTAD (Mishra et al., 2021), MPDD (Jezek et al., 2022), KSDD (Tabernik et al., 2020), and KSDD2 (Božič et al., 2021), are more closely associated with specific industrial inspection scenarios and typically involve more constrained object categories or acquisition conditions. MVTec LOCO-AD (Bergmann et al., 2022a) extends this setting by including logical anomalies in addition to structural anomalies, while Real-IAD (Wang et al., 2024) introduces large-scale, high-resolution, and multi-view data collected from real production lines. The datasets also differ in sensing modality and construction strategy. MVTec 3D-AD (Bergmann et al., 2022b) focuses on industrial 3D sensing, whereas Eyecandies (Bonfiglioli et al., 2022) provides a synthetic multimodal setting with RGB, depth, and normal maps. In contrast, datasets such as DAGM (Wieler et al., 2007) and DTD-Synthetic (Aota et al., 2023) are fully synthetic and mainly used for controlled or texture-oriented evaluation. Mixed datasets, including PAD (Zhou et al., 2023) and ISP-AD (Krassnig & Gruber, 2025), combine real and synthetic components to balance controllability

and realism. Overall, these variations imply that IAS methods may be evaluated under different assumptions depending on the selected benchmark, including differences in visual appearance, anomaly type, and supervision setting.

Table 4: Key metrics commonly used in IAS evaluation.

| Metric | Level | Formula | Remarks/usage |
|---|---|---|---|
| **Synthesis / Editing Metrics** | | | |
| FID | $\downarrow$ | $\|\mu_r - \mu_g\|_2^2 + \mathrm{Tr}\left(\Sigma_r + \Sigma_g - 2(\Sigma_r \Sigma_g)^{1/2}\right)$ | Distributional realism between real and generated features (often computed on full images or cropped regions / patches). |
| Reference-based LPIPS | $\downarrow$ | $d(x, y)$ | Perceptual distance between a generated or edited image and a reference image; used for perceptual fidelity, editing consistency, or background preservation. |
| IC-LPIPS | $\uparrow$ | $\frac{1}{K}\sum_{k=1}^{K}\frac{2}{n_k(n_k-1)}\sum_{i<j}^{n_k} d(x_i^{(k)}, x_j^{(k)})$ | Intra-cluster LPIPS used for anomaly diversity in recent IAS evaluations. Generated anomalies are grouped by real anomaly references or clusters, and higher intra-cluster pairwise LPIPS indicates greater diversity. |
| PSNR | $\uparrow$ | $10\log_{10}\left(\mathrm{MAX}^2/\mathrm{MSE}\right)$ | Pixel-level reconstruction fidelity; often used to assess background preservation in local editing. |
| SSIM | $\uparrow$ | $\dfrac{(2\mu_x\mu_y + c_1)(2\sigma_{xy} + c_2)}{(\mu_x^2 + \mu_y^2 + c_1)(\sigma_x^2 + \sigma_y^2 + c_2)}$ | Structure preservation under synthesis or editing. |
| Alignment IoU | $\uparrow$ | $\dfrac{|\hat{M} \cap M|}{|\hat{M} \cup M|}$ | Mask-conditioned control fidelity (generated / edited region vs. target mask). |
| **Downstream Detection Metrics** | | | |
| Precision (P) | $\uparrow$ | $\dfrac{TP}{TP + FP}$ | Controls false alarms; proportion of predicted anomalies that are correct. |
| Recall (R) | $\uparrow$ | $\dfrac{TP}{TP + FN}$ | Coverage of true anomalies; avoids missed anomalies. |
| FPR | $\downarrow$ | $\dfrac{FP}{FP + TN}$ | False alarm rate on normal samples; often used to trace ROC / PRO-style curves. |
| Image-level AUROC | $\uparrow$ | $\int_0^1 \mathrm{TPR}(u)\,du, \quad u = \mathrm{FPR}$ | Threshold-free image-level anomaly detection performance. |
| Image-level AP | $\uparrow$ | $\sum_n (R_n - R_{n-1})P_n$ | Preferred under class imbalance; summarizes the precision–recall curve without linear interpolation. |
| F1 score | $\uparrow$ | $\dfrac{2PR}{P + R}$ | Balances precision and recall; overall image-level detection effectiveness. |
| **Localization / Segmentation Metrics** | | | |
| Pixel-level AUROC | $\uparrow$ | $\int_0^1 \mathrm{TPR}(u)\,du, \quad u = \mathrm{FPR}$ | Pixel-wise localization quality (anomaly map vs. GT mask). |
| PRO | $\uparrow$ | $\dfrac{1}{|\mathcal{C}|}\sum_{C \in \mathcal{C}}\dfrac{|\hat{M}_\tau \cap C|}{|C|}$ | Per-region overlap at threshold $\tau$; averages region-wise recall over connected GT components. |
| AU-PRO | $\uparrow$ | $\int_0^\alpha \mathrm{PRO}(u)\,du, \quad u = \mathrm{FPR}$ | Area under the PRO curve; commonly reported with an FPR limit $\alpha$ (often 0.3). |
| sPRO | $\uparrow$ | $\dfrac{1}{|\mathcal{C}|}\sum_{C \in \mathcal{C}} \min\left(\dfrac{|\hat{M}_\tau \cap C|}{s(C)}, 1\right)$ | Saturated PRO for settings with logical / weakly pixel-defined anomalies; $s(C)$ is a dataset-defined saturation threshold for region $C$. |
| AU-sPRO | $\uparrow$ | $\int_0^\beta \mathrm{sPRO}(u)\,du, \quad u = \mathrm{FPR}$ | Area under the sPRO curve; commonly used in MVTec LOCO-style evaluation, often with $\beta = 0.05$. |
| IoU | $\uparrow$ | $\dfrac{|\hat{M} \cap M|}{|\hat{M} \cup M|}$ | Thresholded mask overlap for segmentation / localization. |
| mIoU | $\uparrow$ | $\dfrac{1}{N}\sum_{i=1}^{N}\dfrac{|\hat{M}_i \cap M_i|}{|\hat{M}_i \cup M_i|}$ | Mean IoU over samples. |
| Dice (mask-F1) | $\uparrow$ | $\dfrac{2|\hat{M} \cap M|}{|\hat{M}| + |M|}$ | Segmentation overlap; often more sensitive to small anomalous regions than IoU. |

*Note: $TP, FP, TN, FN$ denote true / false positives / negatives; $\hat{M}$ and $M$ denote predicted and ground-truth masks; $\hat{M}_\tau$ denotes the thresholded predicted mask at threshold $\tau$; $\mathcal{C}$ is the set of connected components in the ground-truth anomaly mask; $s(C)$ denotes the saturation threshold associated with region $C$; $\alpha$ and $\beta$ are the integration limits used for AU-PRO and AU-sPRO, respectively.*

Metric selection is therefore not a neutral reporting detail. Some studies mainly test whether synthesized anomalies improve downstream detectors, whereas others further examine visual realism, structural preservation, editing faithfulness, or image–mask alignment. Table 4 organizes these criteria into synthesis/editing, downstream detection, and localization/segmentation metrics, which correspond to different technical claims in IAS evaluation.

Metric design should also be interpreted under the same benchmark logic. As the dataset family, anomaly form, and supervision granularity change, the role of evaluation metrics changes accordingly. This distinction is important because realism, detector improvement, and spatial accuracy do not correspond to the same notion of synthesis quality. Accordingly, IAS evaluation can be understood from three complementary perspectives: direct synthesis quality, downstream detection utility, and spatial localization accuracy.

The first perspective concerns direct synthesis quality. Metrics such as FID and LPIPS are most informative when the generated image is directly used as an evaluation target rather than merely as a training artifact. In this setting, the question is not only whether the synthesized anomaly helps detection, but also whether it is visually plausible and statistically close to real abnormal data. FID is commonly used as a coarse feature-space realism indicator. LPIPS should be interpreted according to its computation protocol: reference-based LPIPS is lower-is-better for perceptual fidelity, whereas IC-LPIPS is higher-is-better for anomaly diversity in recent IAS evaluations (Sun et al., 2025; Hu et al., 2024b; Zhang et al., 2024). When the task involves local editing rather than full-image generation, PSNR and SSIM become more meaningful because they reflect background preservation and structural continuity outside the manipulated region. Alignment IoU is particularly relevant in mask-guided settings, where a method is expected to place the anomaly in the intended region with sufficient spatial controllability.

The second perspective concerns downstream detection utility. IAS is rarely judged by visual quality alone, because many studies use synthesized anomalies primarily to improve anomaly detection through more effective training data. Image-level AUROC remains one of the most widely used summary metrics because it is threshold-free and relatively easy to compare across methods. However, AUROC alone may be insufficient in industrial settings with low anomaly prevalence and substantial class imbalance. Under such conditions, image-level AP is often more informative because it captures the precision–recall trade-off more directly, while F1 score is useful for reporting balanced performance at a selected operating point. Precision and recall further help clarify whether a reported gain mainly comes from better anomaly coverage, fewer false alarms, or a compromise between the two. Together, these metrics address a central practical question in IAS: whether synthesized anomalies genuinely improve downstream discrimination rather than merely producing visually appealing abnormal samples.

The third perspective concerns spatial localization accuracy. Once a method claims stronger spatial correspondence, region-aware supervision, or improved local controllability, localization and segmentation metrics become necessary. Pixel-level AUROC is a common starting point because it evaluates anomaly maps in a threshold-free manner, but pixel-wise ranking alone does not fully characterize localization quality in industrial settings. PRO and AU-PRO are therefore particularly important, since they assess overlap at the connected-region level and are often better aligned with practical anomaly localization than raw pixel accuracy alone. IoU, mIoU, and Dice are more directly tied to explicit mask agreement and are especially useful for local synthesis, mask-guided editing, or paired supervision. Dice is often more sensitive to small abnormal regions, whereas IoU imposes a stricter overlap criterion. In settings involving logical anomalies or less sharply defined abnormal regions, sPRO and AU-sPRO become more appropriate because they better match the evaluation protocols used in such settings.

These three perspectives are not equally emphasized across IAS paradigms. For hand-crafted and distribution hypothesis-based methods, synthesized anomalies often function mainly as intermediate training resources, so the most direct evidence of usefulness usually comes from downstream detection gains. By contrast, GM-based and VLM-based methods often make stronger claims about realism, controllability, image–mask consistency, or prompt-guided editing quality, which cannot be supported by AUROC or AP alone and therefore require direct synthesis metrics and alignment-oriented measures as well. When such methods are further used to provide dense supervision or spatially controllable augmentation, localization metrics are needed to verify that improved realism does not come at the expense of accurate region correspondence.

Metric choice is therefore tightly coupled with benchmark choice. On standard real 2D datasets such as MVTec-AD and VisA, image-level AUROC, pixel-level AUROC, AU-PRO, IoU, and Dice often provide a sufficiently solid picture of whether synthesized anomalies improve detector training and localization quality. On MVTec LOCO-AD, protocol-aware measures such as sPRO or AU-sPRO become more suitable because the evaluation target includes logical inconsistency in addition to ordinary abnormal regions. In local editing or controllable generation settings, metrics such as SSIM, PSNR, reference-based LPIPS, IC-LPIPS, and Alignment IoU carry greater weight because they jointly reflect whether the generated anomaly is visually plausible, perceptually consistent, diverse, and spatially well grounded. Metric selection in IAS should therefore be understood as part of the technical claim itself rather than as a cosmetic reporting choice.

Taken together, Table 4 provides a compact view of how IAS methods are evaluated in practice. Direct synthesis metrics indicate whether the generated anomaly is visually plausible and aligned with the intended control signal; downstream detection metrics indicate whether the synthetic data improves discriminative learning; and localization metrics indicate whether this benefit extends to spatially precise anomaly identification. Read together with Table 3, this metric view establishes the evaluation framework for the cross-paradigm review that follows.

## 3 Hand-crafted Synthesis

As illustrated in Fig. 4, hand-crafted synthesis constructs samples with anomalies through predefined operations on normal images. Compared with later paradigms that rely on learned generative priors, this family is training-free, simple to implement, and easy to deploy in low-resource settings. Its central idea is to create anomalies by manually altering image content or visual completeness, and the main differences among subcategories lie in whether the anomaly is derived from the image itself, introduced from external sources, or formed by deliberately disrupting local structure.

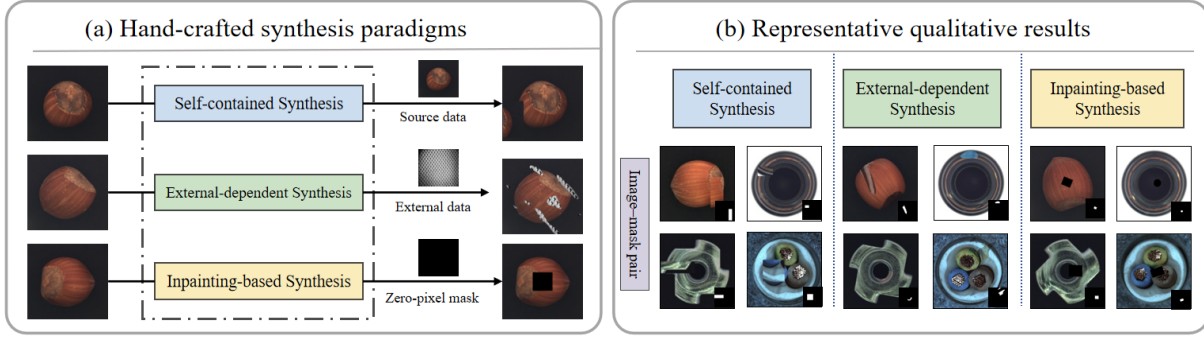

Figure 4: Hand-crafted synthesis. (a) Overview of three representative hand-crafted pipelines, including self-contained synthesis, external-dependent synthesis, and inpainting-based synthesis. These methods synthesize anomaly content through rule-based image manipulation without training a dedicated generative model. (b) Representative qualitative results adapted from (Xu et al., 2025b;c), including examples of methods such as CutPaste (Li et al., 2021) and DRAEM (Zavrtanik et al., 2021a).

**Self-contained synthesis** represents the most direct form of hand-crafted IAS, in which abnormal regions are constructed from available normal images without introducing auxiliary external texture sources. Typical operations include cropping, copying, rearranging, and local perturbation. Early representative methods such as CutPaste (Li et al., 2021) crop local patches and paste them into normal images, while NSA (Schlüter et al., 2022) blends scaled patches sampled from separate normal images using Poisson image editing. REB (Lyu et al., 2024) introduces Bézier-curve-based region design to obtain more flexible and irregular anomaly shapes. Other methods further extend this line by using traditional augmentation, local copying, or region-level transformation to perturb normal images more adaptively, as in ADSAS (Pei et al., 2023), ADAFF (Zhang et al., 2023a), VDDM (Leyendecker et al., 2023), and FEGAN (Fan et al., 2024). ADDLA (Yoa et al., 2021) further combines multiple local transformations dynamically, making the synthesis

process less rigid than simple crop-and-paste designs. By reusing content from normal data, these methods can construct pseudo-anomalies without relying on auxiliary texture libraries, while retaining a lightweight and controllable synthesis process. This makes them particularly suitable for rapid and lightweight augmentation. However, because the synthesized anomaly characteristics are determined by predefined operations, shapes, and transformation strategies, there is no explicit guarantee that they will faithfully reproduce complex real anomalies. Consequently, this subcategory is less suitable when downstream deployment requires high-fidelity reproduction of complex real anomalies whose morphology and appearance cannot be adequately represented by the chosen perturbations.

**External-dependent synthesis** builds on the previous idea by introducing anomaly content that does not originate from the source image itself. Instead of relying only on internal structures, it blends external textures or auxiliary visual patterns into normal images, thereby increasing appearance diversity. Representative methods such as DRAEM (Zavrtanik et al., 2021a) and DeSTSeg (Zhang et al., 2023b) use Perlin-noise-based masks to combine clean backgrounds with external textures, while MemSeg (Yang et al., 2023) further constrains the anomaly region to the foreground to reduce semantic mismatch between synthesized anomalies and object regions. More recently, CAI (Wang et al., 2026) extends this principle beyond generic texture sources by transferring real anomaly patterns from other industrial domains and injecting them into target normal images through scale-aware matching, foreground-constrained placement, and Poisson blending. Relative to self-contained synthesis, this subcategory broadens the anomaly space by allowing content that is unavailable in the original image. Its main advantage is that it can generate richer appearance variation than purely internal manipulation, making it particularly suitable for augmentation scenarios that require broad appearance variation. At the same time, once anomaly content is imported from external sources, local compatibility becomes more difficult to maintain. Blending artifacts, boundary inconsistency, and semantic mismatch may therefore appear, especially when the inserted content is visually plausible in isolation but poorly aligned with the surrounding industrial context. Consequently, this subcategory is less suitable for applications that require strict local semantic and boundary consistency between synthesized anomalies and their surrounding context.

**Inpainting-based synthesis** differs from the previous two subcategories by creating anomalies through masking selected regions and intentionally breaking local visual completeness. In this sense, it can be viewed as a complementary hand-crafted strategy: whereas self-contained and external-dependent methods mainly create "added" abnormal content, inpainting-based methods more often simulate missing or corrupted local patterns. This makes them particularly compatible with reconstruction-oriented anomaly detection. Typical methods such as I3AD (Nakanishi et al., 2021), RIAD (Zavrtanik et al., 2021b), and SMAI (Li et al., 2020) generate pseudo-anomalies by masking random regions and reconstructing them from surrounding information. Later variants improve this basic recipe by making masking more adaptive or structured. For example, InTra (Pirnay & Chai, 2022) introduces transformer-based inpainting for anomaly-oriented reconstruction, and AMI-Net (Luo et al., 2024) develops an adaptive mask generator that preserves surrounding normal background while selectively concealing target regions. Other methods, including CDO (Cao et al., 2023), RD4AD (Tien et al., 2023), and CACL (Jin & Heizmann, 2024), further explore masking-based perturbations through random noise injection, cutout-style black masks, or related corruption strategies. By directly constructing incomplete or corrupted local patterns, this subcategory is especially suitable for reconstruction-oriented pipelines and usually remains easy to control spatially. However, its synthesized anomaly space is primarily shaped by masking and corruption operations. Consequently, this subcategory is less suitable when target anomalies cannot be adequately represented by missing or corrupted local patterns, particularly when their appearance depends on more complex morphology or domain-specific visual characteristics. Table 5 summarizes the practical characteristics of hand-crafted IAS from an engineering perspective, with particular emphasis on resource burden, deployment suitability, and practical limitations. Since these methods are rule-based, their training cost is marked as N/A across all three subcategories, and inference overhead is generally low because synthesis mainly relies on patch manipulation, mask sampling, blending, or simple local corruption rather than iterative generation. The main runtime difference lies in memory usage: self-contained and inpainting-based methods typically require only lightweight buffers, whereas external-dependent synthesis may additionally maintain texture banks or reference sources.

Table 5: Practical profile of hand-crafted IAS

| Family | Resource burden | | | Practical ceiling | | Deployment / limitation analysis | |
|---|---|---|---|---|---|---|---|
| | Train | Infer | Memory | Control | Realism | Suitable scenario | Practical limitation |
| Self-contained | N/A | Low | Low | Medium | Low | Rapid lightweight augmentation | Limited diversity and synthetic artifacts |
| External-dependent | N/A | Low | Medium | Medium | Medium | Broad appearance diversification | Boundary inconsistency and semantic mismatch |
| Inpainting-based | N/A | Low | Low | Medium | Low | Reconstruction-oriented pipelines | Missing/corrupted appearance bias |

*Note:* Ratings are qualitative and relative to this paradigm; N/A = not applicable. Train/Infer/Memory indicate resource burden; Control/Realism indicate practical ceiling.

Overall, the three subcategories serve different deployment scenarios. Self-contained methods are well suited to rapid and lightweight augmentation, especially when synthetic samples are primarily used to expand training data at low cost; however, they are less suitable when deployment requires high-fidelity reproduction of complex real defects. External-dependent methods are useful when broader appearance variation is needed, because auxiliary textures or visual patterns can enlarge the range of synthesized abnormal content; nevertheless, they are less suitable for applications that require strict local semantic and boundary consistency between the inserted anomaly and the surrounding industrial context. Inpainting-based methods are naturally compatible with reconstruction-oriented pipelines and are particularly appropriate for simulating missing, occluded, or locally corrupted structures; however, they are less suitable when the target defects cannot be adequately represented by such incomplete or corrupted local patterns. The practical limitations in Table 5 further indicate several recurring failure risks: self-contained synthesis may produce limited diversity or artificial artifacts, external-dependent synthesis may suffer from boundary inconsistency and semantic mismatch, and inpainting-based synthesis may bias anomalies toward missing or corrupted appearances. At the paradigm level, hand-crafted synthesis is therefore attractive for rapid deployment, low-resource settings, and lightweight data augmentation, but becomes less suitable as application requirements shift toward faithful reproduction of complex real defects, broader coverage of domain-specific anomaly modes, or strict contextual consistency.

## 4 Distribution Hypothesis-Based Synthesis

As illustrated in Fig. 5, distribution hypothesis-based synthesis constructs abnormal samples or representations by introducing controlled deviations with respect to the distribution of normal data. Compared with hand-crafted synthesis, this paradigm shifts the main emphasis from predefined local pixel manipulations toward distribution-guided perturbation, replacement, or sampling in sample, latent, or feature space. It generally avoids iterative image-space generation and is therefore attractive for anomaly detection and localization when synthetic abnormalities primarily serve as training or evaluation signals rather than visually inspectable outputs. The main distinction within this paradigm lies in whether abnormal deviations are guided by explicit assumptions about the geometry of the normal distribution or constructed without predefined geometry through more direct perturbation and replacement strategies.

**Prior-dependent synthesis** represents the more structured branch of this paradigm, in which the construction, interpretation, or evaluation of abnormal deviations is guided by explicit assumptions about the geometry of the normal distribution. Typical formulations describe normality through a compact manifold, hypersphere, constrained region, or other structured distributional model, and then characterize abnormality according to deviations relative to this geometry. Representative methods include the feature-level GAS branch of GLASS (Chen et al., 2024b), which synthesizes near-distribution abnormal features through Gaussian perturbation guided by gradient ascent and truncated projection under manifold and hypersphere constraints. PBAS (Chen et al., 2024a) learns an approximate boundary of normal features and directionally synthesizes abnormal features at flexible scales under hypersphere guidance, followed by progressive boundary refinement. ADSPR (Shin et al., 2023) constructs Gaussian-perturbed samples and exploits score-based restoration behavior to characterize perturbation resilience around the data manifold. SPVAE (Naud & Lavin, 2020) models visual representations on constant-curvature manifolds through a hyperspherical VAE,

providing a non-Euclidean distributional prior for anomaly-oriented latent organization. By introducing explicit assumptions about normal-distribution geometry, these methods provide a structured reference for constructing, interpreting, or evaluating deviations from normality. For methods that explicitly synthesize abnormal features near the learned normal boundary, such as GLASS and PBAS, this structured geometry can further support boundary-focused discriminative training. At the broader subcategory level, prior-dependent synthesis is particularly suitable for detection settings in which normal representations exhibit a sufficiently stable and structured distribution, such that abnormality can be meaningfully characterized relative to the assumed geometry. However, it is less suitable when a stable geometric description of normality is difficult to justify, because highly heterogeneous normal representations may not be adequately characterized by a compact manifold, hypersphere, or similarly structured prior.

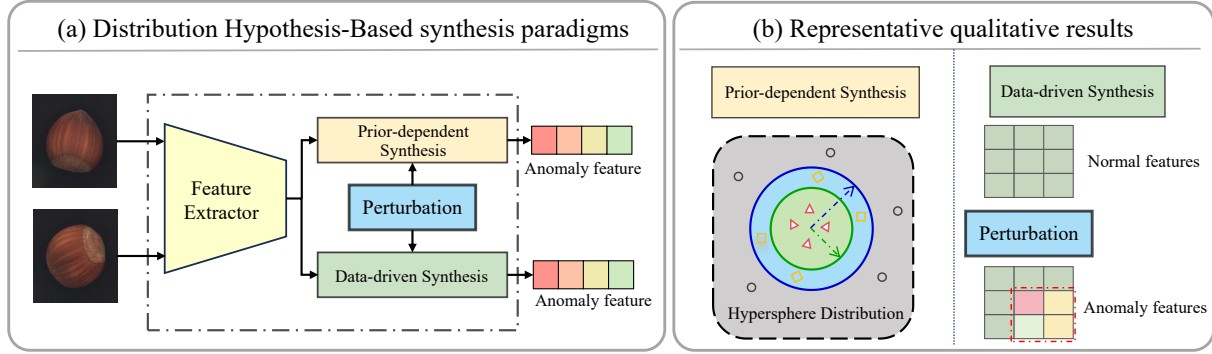

Figure 5: Distribution hypothesis-based synthesis. (a) Overview of the distribution hypothesis-based paradigm. It constructs abnormal samples or representations by introducing controlled deviations around the distribution of normal data, including prior-dependent strategies guided by explicit geometric assumptions and data-driven strategies based on direct perturbation, replacement, or sampling. (b) Representative qualitative illustrations adapted from (Chen et al., 2024b).

**Data-driven synthesis** relaxes explicit geometric assumptions and instead constructs abnormal representations through direct perturbation, jittering, replacement, or sampling of learned features. Rather than defining abnormality relative to a predefined manifold or hypersphere, methods in this subcategory use the learned representation together with a perturbation mechanism to construct abnormal training signals. SimpleNet (Liu et al., 2023), for example, synthesizes abnormal features by adding Gaussian noise to adapted normal features. UniAD (You et al., 2022) introduces feature jittering to disturb input features and encourages the reconstruction model to recover normal information from noisy representations. SuperSimpleNet (Rolih et al., 2025) further localizes feature-space anomaly synthesis by adding Gaussian perturbations within regions specified by binarized Perlin-noise masks. DSR (Zavrtanik et al., 2022) takes a different route through a learned quantized feature space: it generates abnormal representations by replacing masked quantized features with codebook vectors sampled under similarity constraints, thereby encouraging near-distribution anomaly synthesis. Compared with prior-dependent synthesis, this subcategory accommodates a broader range of feature-space perturbation, replacement, and sampling mechanisms without requiring a predefined geometric model of normality. This makes it particularly suitable for anomaly detection and localization pipelines that can directly benefit from feature-level perturbation as synthetic training supervision or regularization. However, because the constructed abnormalities primarily remain in representation space, this subcategory is less suitable for applications that require explicit photorealistic abnormal images, visually inspectable synthetic samples, or reusable image-space anomaly datasets.

Table 6 summarizes the practical characteristics of distribution hypothesis-based IAS from an engineering perspective, with particular emphasis on resource burden, deployment suitability, and practical limitations. Unlike hand-crafted pipelines, these methods require non-trivial training because anomaly synthesis is built on learned distributions or representations, such as encoders, autoencoders, feature extractors, score models, or auxiliary latent components. Their inference overhead nevertheless remains relatively low, since synthesis or perturbation usually involves feature extraction followed by latent perturbation, projection, replacement, sampling, or restoration rather than iterative image-space generation. Memory usage is typically dominated

by the backbone representation model and other additional components such as codebooks or auxiliary heads.

Table 6: Practical profile of distribution hypothesis-based IAS

| Family | Resource burden | | | Practical ceiling | | Deployment / limitation analysis | |
|---|---|---|---|---|---|---|---|
| | Train | Infer | Memory | Control | Realism | Suitable scenario | Practical limitation |
| Prior-dependent | Medium | Low | Medium | Medium | Low | Detection under structured normal distributions | Unstable normal-geometry assumption |
| Data-driven | Medium | Low | Medium | Low | Low | Detection/localization with feature perturbation | Strong reliance on feature quality |

*Note:* Ratings are qualitative and relative to this paradigm. Train/Infer/Memory indicate resource burden; Control/Realism indicate practical ceiling.

Overall, the two subcategories serve distinct deployment scenarios. Prior-dependent methods are well suited to detection settings in which normal representations exhibit a sufficiently stable and structured distribution, allowing abnormal deviations to be constructed, interpreted, or evaluated relative to an explicit geometric reference. They are less suitable when such a stable normal-geometry assumption is difficult to justify. Data-driven methods are particularly suitable for anomaly detection and localization pipelines that can benefit directly from feature-level perturbation, replacement, or sampling without requiring a predefined geometric model of normality. However, they are less suitable when deployment requires explicit photorealistic image-space anomaly generation, visually inspectable synthetic samples, or reusable synthetic image datasets. The practical limitations in Table 6 therefore reflect representation-level failure risks: prior-dependent methods may become unreliable when the assumed normal geometry is unstable, while data-driven methods depend strongly on feature quality and may produce abnormal signals that transfer weakly to real image-space defects. At the paradigm level, distribution hypothesis-based synthesis is therefore attractive when downstream tasks can exploit distribution- or representation-level abnormal signals without requiring direct image-space generation. Its limitations become more apparent as application requirements shift toward visually explicit, photorealistic, and directly reusable synthetic anomalies in image space.

## 5    GM-based Synthesis

As shown in Fig. 6, GM-based synthesis learns image-space priors to produce explicit synthetic images containing anomalies. Compared with hand-crafted synthesis and distribution hypothesis-based synthesis, this paradigm directly models image-level appearance and therefore generally offers a higher ceiling for visual realism and anomaly diversity. These benefits, however, usually come with higher training and inference costs, together with a stronger need to maintain consistency between synthesized anomalies and surrounding normal content. The main distinction within this paradigm lies in whether the model synthesizes a complete image, translates a normal image into the abnormal domain, or edits only designated local regions.

**Full-image synthesis** represents the most direct generative route in this paradigm, because it learns to synthesize samples with anomalies directly from random noise. Typical methods use GANs or diffusion models to approximate the anomaly distribution and then generate visually rich images with anomalies through learned sampling. Early representative work such as Multistage GAN (Liu et al., 2019) decouples texture synthesis from background–anomaly fusion to better model both local anomalies and contextual coherence. Con-GAN (Du et al., 2022) further addresses data scarcity through shared augmentation and a hypersphere-based objective, while DFMGAN (Duan et al., 2023) adopts a two-stage strategy that first learns from normal samples and then fine-tunes on abnormal ones. More recent methods push realism further by incorporating stronger image priors. Defect Spectrum (Yang et al., 2025) combines large receptive fields with patch-level refinement to model global structure and local detail jointly, and RealNet (Zhang et al., 2024) synthesizes more realistic global anomaly content by perturbing variance during reverse diffusion. Because the output is a standalone complete image rather than an edit anchored to a specified source sample, this subcategory is particularly suitable for whole-image anomaly dataset expansion when diverse and realistic synthetic samples are required. However, it is less suitable when deployment requires strict preservation

of a particular source image or precise region-level editing. Moreover, full-image synthesis methods are highly dependent on the quantity and diversity of available training data. When real abnormal samples are scarce, the generated images may become unstable, artifact-prone, or insufficiently aligned with real industrial anomaly morphology.

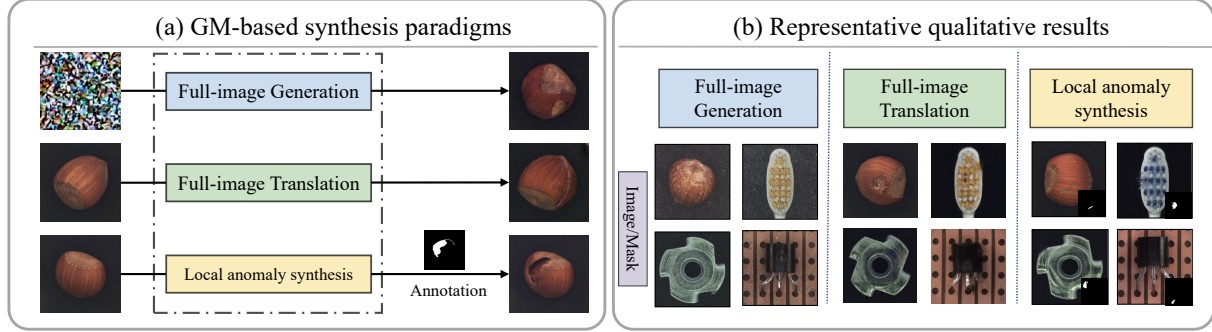

Figure 6: GM-based synthesis. (a) Overview of the GM-based paradigm. It employs deep GMs, such as GANs and diffusion models, for anomaly synthesis, including full-image synthesis, full-image translation, and local anomaly synthesis with region-wise editing. (b) Representative qualitative results adapted from SDGAN (Niu et al., 2020), RealNet (Zhang et al., 2024), and SARD (Wang et al., 2025).

**Full-image translation** starts from a normal image and maps it toward the abnormal domain. Unlike full-image synthesis, the output remains anchored to an existing source image, making this subcategory more suitable for retaining global layout and contextual information during anomaly synthesis. It synthesizes abnormal samples by transforming normal images through domain translation techniques such as CycleGAN (Zhu et al., 2017) and Pix2PixGAN (Isola et al., 2017). Full-image translation learns mappings between normal and abnormal domains, introducing targeted modifications, such as scratches or stains, while preserving the overall image structure. Wen et al. (2022) extend CycleGAN within an anomaly detection pipeline and achieve strong performance. Niu et al. (2020) propose SDGAN, a CycleGAN-based framework that incorporates additional discriminators to refine the translation process. By focusing on modeling the distribution of abnormal samples, SDGAN improves the realism of synthesized abnormal content. Furthermore, Zhang et al. (2021) introduce Defect-GAN, which explicitly models both defacement and restoration processes rather than synthesizing abnormal content in an unconstrained manner, and leverages spatial distribution maps to preserve the appearance of normal backgrounds, thereby improving structural consistency in translated images. Industrial methods such as AttenCGAN (Wen et al., 2022) further adapt translation mechanisms to anomaly-oriented synthesis. Because the synthesis process is conditioned on a normal source image, this subcategory is particularly suitable for normal-to-abnormal transfer when source structure and contextual information should be retained. However, the learned mapping is still shaped by the abnormal domains represented during training, making this subcategory less suitable for open-ended synthesis of anomaly modes that fall substantially outside those learned domains.

**Local anomaly synthesis** restricts synthesis to selected regions while preserving most of the surrounding normal content. Relative to the previous two subcategories, it provides a more targeted mechanism for controlling where abnormal content appears and is therefore naturally aligned with spatially supervised downstream tasks. Representative methods explore mask-guided background preservation, abnormal foreground synthesis, and region-constrained diffusion. For instance, MDGAN (Wei et al., 2022) construct pseudo-normal backgrounds in abnormal regions to emphasize anomaly distributions, DCDGANc (Wei et al., 2023) trains on anomaly-only textures and blends them with different backgrounds for diversified multi-class synthesis, Open-set Fabric (Gao et al., 2025) explicitly encodes anomaly type and mask shape to transfer localized fabric anomalies under region constraints, and SARD (Wang et al., 2025) performs mask-constrained diffusion synthesis that updates only anomaly regions while preserving the surrounding background for segmentation-aware generation. Because abnormal content is restricted to designated regions, this subcategory is particularly suitable for segmentation-oriented and spatially controlled augmentation. It also alleviates one of the major drawbacks of full-image generation, namely the risk that unrealistic global back-

grounds interfere with downstream learning. However, its localized synthesis scope makes it less suitable when target anomalies are diffuse, image-wide, or cannot be meaningfully represented through clear local regions. Moreover, many local editing pipelines rely on masks or other spatial annotations to define where the anomaly should appear, which increases annotation cost and may limit scalability in practical industrial settings.

Table 7: Practical profile of GM-based IAS

| Family | Resource burden | | | Practical ceiling | | Deployment / limitation analysis | |
|---|---|---|---|---|---|---|---|
| | Train | Infer | Memory | Control | Realism | Suitable scenario | Practical limitation |
| Full-image synthesis | Medium | Medium | Medium | Low | High | Whole-image anomaly dataset expansion | Mode collapse and domain overfitting |
| Full-image translation | Medium | Medium | Medium | Low | High | Normal-to-abnormal source translation | Weak spatial controllability |
| Local anomaly synthesis | Medium | High | High | High | High | Spatially controlled segmentation augmentation | Boundary artifacts and mask dependence |

*Note:* Ratings are qualitative and relative to this paradigm. Train/Infer/Memory indicate resource burden; Control/Realism indicate practical ceiling.

Table 7 summarizes the practical characteristics of GM-based IAS from an engineering perspective, with particular emphasis on resource burden, deployment suitability, and practical limitations. Compared with hand-crafted synthesis and representation-space perturbation, GM-based synthesis relies on learned image-space priors and therefore usually entails higher training cost. Its runtime is typically affected by generators, discriminators, encoders, and additional conditioning branches.

Overall, the three subcategories serve distinct deployment scenarios. Full-image synthesis is well suited to whole-image anomaly dataset expansion when standalone, diverse, and realistic synthetic samples are required, but it is less suitable when deployment demands strict preservation of a particular source image or precise region-level editing. Full-image translation is particularly suitable for normal-to-abnormal transfer when source structure and contextual information should be retained, but it is less suitable for open-ended synthesis of anomaly modes that fall substantially outside the learned translation domains. Local anomaly synthesis is naturally suited to segmentation-oriented and spatially controlled augmentation because abnormal content can be restricted to designated regions, but it is less suitable when target anomalies are diffuse, image-wide, or lack clear local support. The practical limitations in Table 7 further highlight image-space failure modes: full-image synthesis may suffer from mode collapse and domain overfitting, full-image translation often provides weak spatial controllability, and local anomaly synthesis may introduce boundary artifacts or depend strongly on mask quality. At the paradigm level, GM-based synthesis is therefore particularly attractive when applications require explicit image-space synthetic samples with learned anomaly appearance, while the choice among its subcategories should depend on whether the deployment objective prioritizes complete-image dataset expansion, source-anchored translation, or localized spatial control.

## 6 VLM-based Synthesis

As summarized in Fig. 7, VLM-based synthesis extends IAS beyond task-specific image-space modeling by exploiting large-scale pre-trained vision–language, multimodal, or text-conditioned generative priors. This distinction is particularly important with respect to the preceding GM-based paradigm. Although both paradigms may adopt similar synthesis architectures, including diffusion models, GM-based methods typically rely on task-specific training, re-training, fine-tuning, or domain adaptation on industrial data, whereas VLM-based methods inherit broadly pre-trained visual and semantic knowledge and transfer it to IAS through prompting, lightweight adaptation, reference conditioning, or multimodal and spatial cues. The boundary between the two paradigms therefore lies primarily in their dependence on large-scale pre-training, knowledge transfer and adaptation strategy, and use of multimodal information, rather than in the choice of a particular synthesis architecture; larger model scale is another secondary characteristic rather than a defining criterion. By leveraging text prompts, reference images, spatial conditions, and other auxiliary cues, VLM-based synthesis further broadens the controllable interface of IAS and enables more explicit specification

of anomaly semantics, appearance, and spatial properties. Within this paradigm, the distinction between single-stage and multi-stage synthesis is determined by the organization of the overall synthesis pipeline: single-stage methods produce the final abnormal output through one unified dominant synthesis process, whereas multi-stage methods explicitly introduce multiple synthesis stages or coordinated sub-pipelines, such as mask synthesis, anomaly-part synthesis, or progressive refinement.

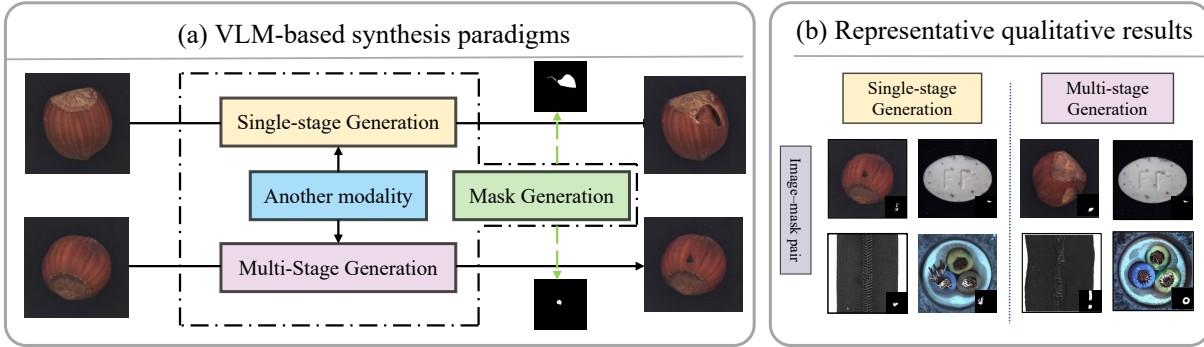

Figure 7: VLM-based synthesis. (a) Overview of the VLM-based paradigm. It leverages large-scale pre-trained vision–language or multimodal diffusion backbones together with text prompts, reference images, spatial conditions, and other auxiliary cues for anomaly synthesis. Single-stage methods produce the final abnormal output through one unified dominant synthesis process, whereas multi-stage methods explicitly introduce multiple synthesis stages or coordinated sub-pipelines, such as mask synthesis, anomaly-part synthesis, or progressive refinement. (b) Representative qualitative results adapted from FAST (Xu et al., 2025b), AnomalyDiffusion (Hu et al., 2024b), and SeaS (Dai et al., 2025).

**Single-stage synthesis** integrates multimodal conditions into one unified dominant synthesis process that directly produces the final abnormal output. Text prompts, reference images, spatial conditions, or predefined masks may be used as inputs, but the workflow does not rely on a separately synthesized intermediate output to drive a subsequent synthesis stage. Therefore, anomaly content is synthesized without introducing an explicitly separated refinement pipeline. Representative methods such as CUT (Sun et al., 2024) and AnomalyAny (Sun et al., 2025) exploit attention or prompt guidance on pre-trained diffusion backbones, DefectFill (Song et al., 2025) uses LoRA fine-tuning to learn anomaly concepts from a few abnormal image–mask pairs and injects them into masked normal images through a single inpainting process, TF-IDG (Xu et al., 2025a) combines retrieval and ControlNet-style conditioning to preserve anomaly shape and detail, and AnoGen (Gui et al., 2024) learns a compact condition embedding for box-guided inpainting under few-shot supervision. Other explicitly multimodal methods, including AnomalyXFusion (Hu et al., 2024a), CAGEN (Jiang et al., 2024), AnomalyControl (He et al., 2024), and FAST (Xu et al., 2025b), further incorporate multimodal semantics, mask-aware control, cross-modal alignment, or foreground guidance while retaining one unified dominant synthesis route. CADiff (Tong et al., 2026) combines text-based anomaly semantics with spatial control in a single diffusion process, ARAS (Qian et al., 2026) edits masked visual tokens under language guidance, CHIMERA (Lee et al., 2026) jointly generates abnormal images and masks within one MM-DiT process, and Anomagic (Jiang et al., 2026) fuses visual and textual prompts to guide a unified inpainting pipeline.

Because the final abnormal output is produced through a relatively direct workflow, this subcategory is particularly suitable for rapid multimodal-conditioned anomaly synthesis when users need to control anomaly semantics or appearance through prompts, references, or available spatial conditions without constructing additional synthesis stages. However, it is less suitable for workflows that require masks, anomaly parts, or other intermediate supervision signals to be separately synthesized and then progressively coordinated or refined. In such settings, a unified synthesis process provides less explicit support for structured multi-output synthesis.

**Multi-stage synthesis** explicitly decomposes anomaly synthesis into multiple synthesis stages or coordinated sub-pipelines. Typical designs first synthesize or model intermediate masks, anomaly parts, coarse

outputs, or auxiliary representations and then use them to guide, align, or refine subsequent synthesis; alternatively, multiple synthesis processes may operate in a coordinated manner. The defining characteristic is therefore not strict sequential execution, but an explicitly decomposed synthesis workflow in which multiple synthesis components jointly contribute to the final output. Representative methods such as DualAnoDiff (Jin et al., 2025) use a dual-branch design to model global context and anomaly regions separately while introducing segmentation-assisted mask generation. DefectDiffu (Shi et al., 2025) combines textual guidance with dedicated mask generation to improve small-anomaly synthesis and more precise control over anomaly properties. GAA (Lu et al., 2025) learns anomaly concept embeddings and then synthesizes semantically aligned anomaly–mask pairs through region-guided mask design, while TDAD (Wei et al., 2025) couples multiscale anomaly synthesis with a two-stage diffusion process to suppress over-reconstruction and recover semantic detail. Other pipelines such as BeltDiff (Zhuang et al., 2025), SeaS (Dai et al., 2025), AnomalyDiffusion (Hu et al., 2024b), AnoStyler (So & Kang, 2026), and AdaBLDM (Li et al., 2024) likewise incorporate self-labeling, token-level attribute binding, textual inversion, or online adaptation to progressively improve mask alignment and local realism. Recently, methods such as AnomalyPainter (Lai et al., 2026) use VLM-generated anomaly descriptions to retrieve relevant textures, which then guide texture-conditioned diffusion synthesis.

Because intermediate outputs and multiple synthesis processes can be explicitly coordinated, this subcategory is particularly suitable for structured synthesis workflows that require synthesized masks, aligned image–mask pairs, separately modeled anomaly parts, or progressive refinement. Such designs are especially useful for segmentation-oriented applications and other downstream tasks requiring spatially aligned supervision. However, the additional synthesis stages, auxiliary branches, and refinement procedures increase computational and engineering complexity. Consequently, this subcategory is less suitable for latency-sensitive or pipeline-simple deployment where synthesis must be completed with minimal runtime and integration overhead.

Table 8 summarizes the practical characteristics of VLM-based IAS from an engineering perspective, with particular emphasis on resource burden, deployment suitability, and practical limitations. Unlike earlier paradigms, the computational burden of this family is largely dominated by large pre-trained backbones, iterative diffusion synthesis, and multimodal conditioning modules. Even when task-specific adaptation is lightweight or partly avoided, runtime memory usage and inference cost may remain substantial because synthesis can still involve large denoising networks, control branches, attention optimization, or additional multimodal encoders.

Table 8: Practical profile of VLM-based IAS

| Family | Resource burden | | | Practical ceiling | | Deployment / limitation analysis | |
|---|---|---|---|---|---|---|---|
| | Train | Infer | Memory | Control | Realism | Suitable scenario | Practical limitation |
| Single-stage VLM-based | Medium | High | High | Medium | High | Multimodal-conditioned anomaly synthesis | Semantic misalignment and weak masks |
| Multi-stage VLM-based | High | High | High | High | High | Structured synthesis with aligned supervision | Error propagation and pipeline complexity |

*Note:* Ratings are qualitative and relative to this paradigm. Train/Infer/Memory indicate resource burden; Control/Realism indicate practical ceiling. Train excludes upstream foundation-model pre-training.

Overall, the two subcategories serve distinct deployment scenarios. Single-stage methods are well suited to multimodal-conditioned anomaly synthesis when users need direct control through prompts, references, or available spatial conditions without constructing additional synthesis stages. However, they are less suitable for workflows that require masks, anomaly parts, or other intermediate supervision signals to be separately synthesized and progressively coordinated or refined. Multi-stage methods are particularly suitable for structured synthesis workflows requiring synthesized masks, aligned image–mask pairs, separately modeled anomaly parts, or progressive refinement, but they are less suitable for latency-sensitive or pipeline-simple deployment. The practical limitations in Table 8 further reflect multimodal and pipeline-level failure risks: single-stage methods may suffer from semantic misalignment or weak masks, while multi-stage methods may propagate errors from intermediate masks, anomaly parts, or retrieved references. At the paradigm level, VLM-based synthesis is therefore particularly attractive when semantic controllability, multimodal

conditioning, and flexible task adaptation are central requirements. Its limitations become more apparent under strict latency and memory budgets, or in highly specialized industrial settings where large-scale pre-trained priors require additional adaptation to represent application-specific anomaly characteristics.

# 7   Practical Guidance and Future Directions

Building on the per-paradigm practical profiles in Tables 5–8, Tables 9 and 10 further provide a cross-paradigm view of IAS subcategories from two complementary perspectives. The former compares their input requirements, synthesis levels, and output forms, whereas the latter summarizes their practical characteristics, including computational cost, controllability, primary target tasks, and key trade-offs. Together, these tables connect the proposed taxonomy with deployment scenarios, annotation and input requirements, downstream tasks, and practical limitations, providing a more diagnostic view of when each paradigm is useful and where it may fail in practice.

Table 9: Input requirements and output forms of IAS subcategories.

| Paradigm | Subcategory | Real abnormal data | Spatial annotation | Text / multimodal prompt | Reference input | Synthesis level | Output form |
|---|---|---|---|---|---|---|---|
| Hand-crafted Synthesis | Self-contained | – | Optional | – | – | Pixel | Abnormal image + mask |
| | External-dependent | – | Optional | – | Required | Pixel | Abnormal image + mask |
| | Inpainting-based | – | Optional | – | – | Pixel | Abnormal image + mask |
| Distribution hypothesis-based Synthesis | Prior-dependent | – | – | – | – | Latent | Abnormal feature |
| | Data-driven | – | – | – | – | Latent | Abnormal feature |
| GM-based Synthesis | Full-image synthesis | Required | Optional | – | – | Latent + pixel | Abnormal image |
| | Full-image translation | Required | Optional | – | – | Latent + pixel | Abnormal image |
| | Local anomaly synthesis | Optional | Optional | – | Optional | Latent + pixel | Abnormal image (+ mask) |
| VLM-based Synthesis | Single-stage | Optional | Optional | Optional | Optional | Latent + pixel | Abnormal image (+ mask) |
| | Multi-stage | Optional | Optional | Optional | Optional | Latent + pixel | Abnormal image + mask |

*Note:* The symbol "–" denotes unused inputs, and "Optional" indicates method-dependent inputs. Spatial annotation refers to external masks, boxes, or pixel labels; reference input refers to external textures, exemplars, or cross-domain anomaly sources. The notation "(+ mask)" means that masks may be available but are not universal.

Several general patterns can be observed. Hand-crafted and distribution hypothesis-based methods are relatively lightweight, but they usually operate within limited input or representation spaces and therefore remain constrained in realism, output richness, or downstream flexibility. GM-based methods improve image-level realism substantially, yet their controllability and practical cost vary considerably across subcategories. VLM-based methods further broaden the interface of IAS by incorporating prompt-based and multimodal conditions, but such gains are often accompanied by higher computational overhead and more complex pipelines. Taken together, these observations highlight that future IAS research should move beyond isolated method design and place greater emphasis on anomaly diversity, controllable synthesis, and multimodal integration.

Improving anomaly diversity remains a central direction. As indicated by Table 9, many existing IAS subcategories either rely on limited anomaly data, operate within relatively restricted synthesis spaces, or produce only feature-level rather than image-level anomaly outputs. Such constraints naturally narrow anomaly coverage and weaken generalization to rare or complex industrial anomalies. This issue is especially evident in methods that depend on scarce real anomaly samples, predefined perturbation rules, or narrowly structured latent assumptions. Future research should therefore develop diversity-oriented synthesis pipelines that can explore underrepresented anomaly modes more effectively. Promising directions include uncertainty-

aware generation, self-supervised discovery of rare anomaly patterns, and active selection of insufficiently covered anomaly modes. Coarse-to-fine synthesis strategies, which first establish global anomaly structure and then refine local details, may also help improve both diversity and realism. More broadly, larger and more diverse industrial anomaly datasets will remain important for reducing overfitting to narrow anomaly distributions and for supporting broader anomaly coverage.

Strengthening controllable anomaly synthesis is equally important. Table 10 reveals a clear controllability gap across IAS subcategories. Although local anomaly synthesis and multi-stage VLM-based methods already provide relatively strong control, many other subcategories still offer only limited control over anomaly type, shape, location, or visual intensity. This weakness directly reduces their value for downstream settings that require accurate spatial supervision, precise image–mask alignment, or fine-grained attribute editing. Future IAS research should therefore place greater emphasis on explicit spatial conditioning, attribute-aware generation, and more structured control over anomaly properties. Large-scale generative models, mask-guided editing frameworks, and segmentation-aware synthesis pipelines are especially promising in this respect, because they can better separate anomaly characteristics from normal background content while preserving structural coherence. More generally, controllability should be treated not as an auxiliary extension, but as a central design objective for downstream-oriented IAS.

Table 10: Deployment-oriented comparison of industrial anomaly synthesis subcategories.

| Paradigm | Subcategory | Cost | Controllability | Downstream tasks | Key trade-off |
|---|---|---|---|---|---|
| Hand-crafted Synthesis | Self-contained | Low | Medium | C/D/S | Good source-background consistency
Limited anomaly diversity |
| | External-dependent | Low | Medium | C/D/S | Richer anomaly appearance
Higher risk of local inconsistency |
| | Inpainting-based | Low | Medium | C/D/S | Simple masking-based synthesis
Anomalies may look missing or corrupted |
| Distribution hypothesis-based Synthesis | Prior-dependent | Medium | Medium | C/D | Clear latent-space structure
Limited flexibility for complex abnormal distributions |
| | Data-driven | Medium | Low | C/D | Adaptive feature perturbation
Strong reliance on feature quality |
| GM-based Synthesis | Full-image synthesis | Medium | Low | C/D | High generation freedom
Weaker preservation of source-image structure |
| | Full-image translation | Medium | Low | C/D | Better domain-level appearance transfer
Weak spatial controllability and limited mask reliability |
| | Local anomaly synthesis | High | High | C/D/S | Strong spatial controllability
Dependence on mask quality and boundary blending |
| VLM-based Synthesis | Single-stage | High | Medium | C/D/S | Flexible prompt-driven synthesis
Weaker spatial precision and mask consistency |
| | Multi-stage | High | High | C/D/S | Better image–mask alignment
Higher pipeline complexity |

*Note:* Ratings are qualitative cross-paradigm comparisons. Cost summarizes relative training/adaptation, inference, and memory burden, excluding upstream foundation-model pre-training. C, D, and S denote image-level anomaly classification, anomaly detection, and pixel-level segmentation, respectively; the listed tasks indicate common downstream support rather than exclusive applicability.

Promoting multimodal anomaly synthesis in a more systematic manner is another important direction. Table 9 shows that multimodal inputs are still concentrated mainly in VLM-based subcategories, whereas most earlier paradigms remain dominated by image-centric or feature-space-oriented synthesis. This suggests that the multimodal potential of IAS has not yet been fully explored. In practical industrial scenarios, however, complementary modalities such as text descriptions, infrared measurements, X-ray scans, depth signals, and process metadata can provide richer semantic and structural cues for anomaly synthesis. Future work should therefore investigate cross-modal alignment strategies that connect such heterogeneous signals with visual anomaly generation more effectively. Potential directions include multimodal transformers, contrastive representation learning, retrieval-augmented generation, and multi-source fusion frameworks that jointly model anomaly appearance, spatial layout, and semantic intent. Progress along this line may enable IAS systems to synthesize anomaly samples that are not only more realistic, but also more controllable and better aligned with practical industrial inspection requirements.

# 8 Conclusion

In this survey, we presented a systematic review of industrial anomaly synthesis (IAS) by examining three central challenges, namely limited anomaly distribution coverage, the difficulty of synthesizing realistic anomaly samples, and the underexplored role of multimodal information. To organize the field in a clearer and more task-oriented manner, we introduced a dedicated taxonomy that groups existing IAS methods into four paradigms: hand-crafted synthesis, distribution hypothesis-based synthesis, GM-based synthesis, and VLM-based synthesis. Based on this taxonomy, we further reviewed representative methods, summarized commonly used datasets and evaluation metrics, and provided deployment-oriented guidance by comparing different IAS subcategories in terms of input requirements, output forms, controllability, cost, downstream tasks, and practical limitations.

Overall, IAS has progressed from heuristic and low-cost perturbation toward more realistic, more controllable, and increasingly multimodal synthesis. Hand-crafted and distribution hypothesis-based methods still provide practical value in lightweight or data-limited settings, whereas GM-based and VLM-based methods have pushed IAS closer to image-level realism and downstream-oriented supervision. At the same time, the field still faces clear limitations in anomaly diversity, fine-grained controllability, and multimodal integration. Continued progress in these directions will be important for developing IAS methods that are not only more realistic, but also more useful for industrial detection, segmentation, and related applications. We hope this survey can serve as a structured reference for understanding existing IAS methods and for supporting future research in this area.

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
