# OpenReview forum: "A Survey on Industrial Anomaly Synthesis"
_TMLR — Under review for TMLR_

### Review · Reviewer_AUfM · 2026-06-14

**Summary Of Contributions:**

This paper presents a dedicated survey on industrial anomaly synthesis (IAS), treating it as an independent research direction rather than an auxiliary component of anomaly detection pipelines.
The authors organize approximately 60 representative methods into four paradigms: hand-crafted synthesis, distribution hypothesis-based synthesis, generative model (GM)-based synthesis, and vision-language model (VLM)-based synthesis.
The survey covers commonly used datasets and evaluation protocols, and provides per-paradigm practical profiles (Tables 4-7) comparing training cost, inference cost, memory footprint, controllability, and realism ceiling.
Future directions around anomaly diversity, controllability, and multimodal integration are discussed.

**Strengths.**
The taxonomy is clearly motivated and consistently applied throughout. The metric selection discussion in Section 2.2 is a genuine contribution: the paper goes beyond listing formulas and explicitly connects metric choice to technical claims, which is practical and non-obvious. The practical profiles in Tables 4-7 give actionable engineering guidance. The scope (60 methods, 17 datasets, 19 metrics) is comprehensive and the paper is well-written.

**Weaknesses.**
Critical analysis of failure modes and deployment boundary conditions is thin: the paper largely describes what each method does but stops short of examining when and why methods fail in practice.
The boundary between GM-based local anomaly synthesis and VLM-based multi-stage generation is not always sharp, and a few methods could plausibly fit either category without the current text making the distinction explicit.

**Audience:**

Yes

**Audience Explanation:**

IAS is growing rapidly as Fig. 1 shows a clear surge in 2025.
Also, practitioners building industrial quality inspection systems lack a structured reference that covers synthesis as a primary topic.
The dataset and metric discussions are useful to researchers entering the area. The cross-paradigm comparison tables (Tables 8-9) are concise and informative.

**Broader Impact Concerns:**

This is a survey paper: the authors review and organize existing methods rather than introducing new synthesis capabilities. The non-intended-use risk is therefore indirect.

That said, IAS methods surveyed here could in principle be misused.
For example, to poison training data for downstream inspection models, or to generate convincing fake defects on conforming products for fraudulent warranty or return claims.

These risks are theoretical and not specific to this paper. A brief acknowledgment would be appropriate if TMLR requires a broader impact statement, but no changes to the technical content are warranted.

**Claims And Evidence:**

Yes

**Claims Explanation:**

This is a survey paper, so the nature of the evidence differs from an empirical contribution.
The primary claims are:
(1) that the four-paradigm taxonomy captures the dominant synthesis mechanisms in the literature;
(2) that different paradigms involve distinct trade-offs in cost, controllability, and realism;
(3) that metric and benchmark selection are not neutral choices but are coupled to the technical claims a method makes.


All three are supported at the appropriate level for a survey.
Claim (1) is substantiated by systematic coverage of ~60 methods across Sections 3-6, with each method mapped consistently to the taxonomy.
Claim (2) is documented in Tables 4-7, which translate the per-method analyses into comparable practical profiles.
Claim (3) is argued in Section 2.2 with explicit reasoning about why certain metrics (e.g., AU-PRO vs. pixel-level AUROC, SSIM vs. FID) are appropriate or insufficient in specific settings: this is the most analytically original part of the paper.

No claim is left at the level of assertion without grounding in the reviewed literature.

**Requested Changes:**

**Major**

DualAnoDiff citation must be updated. The paper cites DualAnoDiff as "arXiv preprint arXiv:2408.13509, 2024." This work was accepted at CVPR 2025 and is available at the CVF Open Access repository (Jin et al., Proceedings of the IEEE/CVF Conference on Computer Vision and Pattern Recognition, 2025, pp. 30420-30429). The citation and any corresponding text in Section 6 should reflect the published version.

**Minor**

1. Add critical analysis of failure modes. The paradigm-level discussion is descriptive rather than diagnostic. A dedicated subsection or paragraph per paradigm discussing known failure cases would substantially strengthen the practical value of the review: boundary artifacts in local synthesis, semantic misalignment in VLM-based methods, mode collapse in GAN-based full-image synthesis.

2. Tie the taxonomy to practical deployment scenarios. Tables 8-9 are a step in this direction, but the paper would benefit from clearer guidance on which paradigm suits which combination of annotation budget, available abnormal data, and downstream task. This would move the survey from descriptive to prescriptive.

3. Clarify the GM/VLM boundary. The distinction between GM-based local anomaly synthesis and VLM-based multi-stage generation should be stated more explicitly. Both operate in image space, both can use diffusion models, and both can produce image-mask pairs. The current text implies the boundary lies in whether a large pretrained vision-language backbone is used, but this is not stated directly. One or two sentences making this explicit would prevent ambiguity.

4. Newly published IAS methods. AAAI 2026 included several IAS papers. These likely postdate the submission, but a brief note acknowledging them as concurrent or subsequent work, rather than leaving the impression the survey is exhaustive through 2026, would improve transparency.

---

> ### Author Response · Authors · 2026-07-09
> **Response to Reviewer AUfM (Part 1/2)**
>
> We thank the reviewer for the thoughtful and detailed review. We are encouraged that the reviewer finds the taxonomy, metric discussion, and practical profiles useful. We agree that the original manuscript should more explicitly discuss failure modes, deployment boundary conditions, the GM/VLM boundary, and recent and concurrent IAS works. We have revised the manuscript accordingly.
>
> **DualAnoDiff citation.**
> We thank the reviewer for pointing out the outdated DualAnoDiff citation. We have updated the citation throughout the manuscript. In Figure 3, Section 6, and the References, DualAnoDiff is now cited as Jin et al. (2025), and the reference entry has been updated to the CVPR 2025 version: "Dual-interrelated diffusion model for few-shot anomaly image generation," Proceedings of the IEEE/CVF Conference on Computer Vision and Pattern Recognition (CVPR), pp. 30420--30429, June 2025.
>
> **Failure modes and deployment boundary conditions.**
> We agree that the original version was more descriptive than diagnostic. We therefore revised Sections 3--6 and Tables 5--8 to add explicit suitable scenarios, practical limitations, and failure modes for each IAS subcategory.
>
> (1) Hand-crafted synthesis: We now discuss limited diversity and synthetic artifacts for self-contained methods, boundary inconsistency and semantic mismatch for external-dependent methods, and bias toward missing or corrupted appearances for inpainting-based methods. We also clarify the corresponding deployment scenarios: rapid lightweight augmentation, broad appearance diversification, and reconstruction-oriented pipelines.
>
> (2) Distribution hypothesis-based synthesis: We now emphasize unstable normal-geometry assumptions for prior-dependent methods and strong reliance on feature quality for data-driven methods. The revised text also explains that this paradigm is suitable when downstream tasks can exploit feature-level abnormal signals, but less suitable when applications require photorealistic image-space anomaly datasets.
>
> (3) GM-based synthesis: We now explicitly discuss mode collapse and domain overfitting in full-image synthesis, weak spatial controllability in full-image translation, and boundary artifacts or mask dependence in local anomaly synthesis. The revised text also distinguishes complete-image dataset expansion, source-anchored translation, and localized spatial control as different deployment objectives.
>
> (4) VLM-based synthesis: We now discuss semantic misalignment and weak masks in single-stage methods, as well as error propagation and pipeline complexity in multi-stage methods. We further clarify that multi-stage methods are useful for structured workflows requiring aligned image-mask pairs, separately modeled anomaly parts, or progressive refinement, but less suitable for latency-sensitive settings or deployments that require simple pipelines.
>
> These changes jointly address Reviewer AUfM's request for failure-mode analysis and Reviewer iR9j's related request for more critical comparisons of assumptions, limitations, and suitable application scenarios. We believe the revised manuscript is now more useful as a deployment-oriented survey rather than merely a descriptive review.
>
> **Connecting the taxonomy to practical deployment scenarios.**
> We have expanded the deployment-oriented discussion in the paper. Section 7 has been retitled from "Future Directions" to "Practical Guidance and Future Directions," and its opening explains how the per-paradigm practical profiles in Tables 5--8 are connected to the cross-paradigm comparisons in Tables 9--10. Together, these tables compare input requirements, synthesis levels, output forms, cost, controllability, downstream tasks, and key trade-offs.
>
> We also revised Table 9 to distinguish real abnormal data, spatial annotation, text or multimodal prompts, and reference inputs. Instead of compact symbols, the table now uses clearer entries such as "Required", "Optional", and "--". The notes also define spatial annotation, reference input, and the notation "(+ mask)". Table 10 now gives a more explicit deployment-oriented comparison of cost, controllability, downstream tasks, and key trade-offs.
>
> Together, these changes help readers choose an IAS paradigm based on practical factors such as annotation budget, availability of real abnormal data, spatial-control needs, multimodal prompting, and downstream targets such as classification, detection, or segmentation.
>
> We also updated the Abstract and Conclusion to reflect this practical-guidance contribution. The revised Abstract now states that the survey provides deployment-oriented comparisons and practical guidance by analyzing input requirements, output forms, controllability, cost, downstream tasks, and practical limitations across IAS subcategories.

---

> ### Author Response · Authors · 2026-07-09
> **Response to Reviewer AUfM (Part 2/2)**
>
> **Clarifying the GM/VLM boundary.**
> We fully agree that the distinction between GM-based local anomaly synthesis and VLM-based multi-stage generation should be stated more explicitly. This issue was also raised by Reviewer iR9j, and we revised Section 2.1, Table 2, and Section 6 to address both reviewers' concerns.
>
> In Section 2.1, we now state that the taxonomy is not based merely on a backbone or synthesis technique. Instead, it distinguishes paradigms by dominant synthesis mechanism, knowledge source, dependence on large-scale pre-training, task adaptation strategy, and modality. We added Table 2 to compare the four paradigms along these criteria.
>
> In Section 6, we now explicitly state that architectural overlap alone does not determine the boundary between GM-based and VLM-based synthesis. The two paradigms may both employ diffusion models, perform local editing, or generate image-mask pairs. Instead, the distinction follows the dominant synthesis paradigm described in Section 2.1, particularly the primary source of synthesis capability, dependence on large-scale pre-training, knowledge-transfer and adaptation strategy, and modality. GM-based methods are characterized mainly by task-specific generative modeling and adaptation for industrial anomaly synthesis, whereas VLM-based methods place large-scale pre-trained vision-language, multimodal, or text-conditioned priors at the center of the synthesis or control process and transfer their semantic or multimodal knowledge to IAS through prompting, lightweight adaptation, reference conditioning, or other multimodal and spatial cues.
>
> We also refined the single-stage/multi-stage distinction within the VLM-based paradigm. Single-stage methods produce the final abnormal output through one unified dominant synthesis pipeline. Multiple prompts, references, masks, or spatial conditions may be used as inputs, but no independently synthesized intermediate output is required to drive subsequent synthesis. In contrast, multi-stage methods explicitly decompose anomaly synthesis into multiple synthesis stages or coordinated sub-pipelines, where synthesized intermediate outputs, such as masks, anomaly parts, coarse results, or auxiliary representations, participate in subsequent or coupled synthesis. This clarification reduces ambiguity for methods that combine diffusion backbones with prompting, masks, reference conditioning, or local editing, thereby addressing both Reviewer AUfM's and Reviewer iR9j's concerns about taxonomy boundaries.
>
> **Recent and concurrent IAS methods.**
> We have updated the taxonomy, method discussion, and references to better reflect recent and concurrent works. The revised literature corpus now covers approximately 64 representative IAS methods. In the Introduction, we explicitly state that the survey focuses on works available from 2019 to March 30, 2026, and that, because IAS is rapidly evolving, especially with concurrent 2026 works, the survey is representative rather than exhaustive.
>
> We also updated Figure 3, Section 6, and the References to include recent and concurrent IAS methods. In particular, we added six recent AAAI 2026 IAS-related works, including Anomagic, ARAS, CADiff, CHIMERA, AnomalyPainter, and AnoStyler, to the taxonomy, method discussion, and References. We also added STAGE and CAI as additional recent and concurrent IAS-related works. These additions acknowledge recent progress without implying that the survey is exhaustive through all of 2026.
>
> **Broader impact point.**
> We agree with the reviewer that the risk of unintended use is indirect because this paper is a survey and does not introduce a new anomaly synthesis capability. We also agree that, in principle, the IAS methods reviewed in this survey could be misused, for example, for training-data poisoning or for generating misleading synthetic defects. Since the reviewer notes that these risks are theoretical and not specific to this paper, and that no changes to the technical content are warranted, we have not added a dedicated broader-impact section in the current revision. We will add a brief acknowledgement if it is requested later in the TMLR process.
>
> A possible concise acknowledgement would be: "Although this survey does not introduce a new anomaly synthesis method, the reviewed IAS techniques may have indirect unintended-use risks, such as poisoning training data for downstream inspection systems or generating misleading synthetic defects. Practical deployment should therefore consider data provenance, domain-expert validation, and downstream robustness checks."
>
> We thank the reviewer again for the detailed suggestions. They helped us improve the survey from a primarily descriptive review into a more diagnostic and deployment-oriented reference.

---

### Review · Reviewer_78DH · 2026-06-15

**Summary Of Contributions:**

This paper surveys industrial anomaly synthesis (IAS), a topic that has become increasingly relevant for industrial anomaly detection. The paper argues that existing industrial anomaly surveys are mostly detection-oriented and do not treat anomaly synthesis as an independent methodological problem. To address this gap, the paper organizes representative IAS methods into four paradigms: hand-crafted synthesis, distribution hypothesis-based synthesis, generative-model-based synthesis, and vision-language-model-based synthesis. It further summarizes commonly used datasets, evaluation metrics, practical trade-offs, and future research directions.

The topic is timely and potentially useful. It well organizes the existing research on anomaly synthesis, and I believe this paper could become a valuable survey.

**Audience:**

Yes

**Audience Explanation:**

Anomaly synthesis has recently garnered extensive research interest, but there is a lack of related surveys that systematically organize existing methods. This paper fills this gap, and I believe it has great potential.

**Broader Impact Concerns:**

No ethical issues.

**Claims And Evidence:**

Yes

**Claims Explanation:**

This paper is a research survey that systematically reviews studies related to anomaly synthesis, scientifically categorizing these methods into four classes. It discusses in detail the advantages and key limitations of each category and provides future research prospects, potentially driving the development of this field.

**Requested Changes:**

I have carefully reviewed the paper and found no major issues or errors. However, there is one minor concern I would like to point out: the LPIPS metric is typically used in the anomaly synthesis community to evaluate the diversity of synthesized anomalies. The standard calculation process involves clustering the synthesized anomalies based on real ones and then measuring the LPIPS score within each cluster, where a larger LPIPS indicates greater diversity in the synthesized anomalies (See AnomalyAny Table 1, AnomalyDiffusion Table 1, RealNet Table S6). I suggest the authors appropriately revise the discussion on the LPIPS metric (Table 3) to highlight how its interpretation differs from that in the general image synthesis domain.

---

> ### Author Response · Authors · 2026-07-09
> **Response to Reviewer 78DH**
>
> We thank the reviewer for the positive assessment and for pointing out the
> IAS-specific interpretation of LPIPS. We agree that the original wording in
> Table 3 was too generic. In the anomaly synthesis literature, LPIPS serves two
> different roles: it can measure perceptual fidelity when computed against a
> reference image, and it can measure intra-cluster diversity when computed among
> synthesized anomalies grouped by real anomaly references or clusters. We have
> revised Table 4 and the corresponding discussion in Section 2.2 accordingly.
>
> **Revision to LPIPS in the metric table.**
> In the revised manuscript, the previous single "LPIPS ↓" entry has been replaced
> by two separate entries in Table 4:
>
> (1) Reference-based LPIPS ↓: This metric measures the perceptual distance
> between a generated or edited image and a reference image. Lower values indicate
> better perceptual fidelity, editing consistency, or background preservation.
>
> (2) IC-LPIPS ↑: This metric measures intra-cluster LPIPS and is used to evaluate
> anomaly diversity. We added the corresponding formula and explicitly state that
> generated anomalies are grouped by real anomaly references or clusters. Higher
> intra-cluster pairwise LPIPS indicates greater diversity.
>
> **Revision to the metric discussion.**
> We also revised the explanatory text in Section 2.2. The revised manuscript now
> explicitly states that LPIPS should be interpreted according to its computation
> protocol: for reference-based LPIPS, lower values indicate better perceptual
> fidelity, whereas for IC-LPIPS, higher values indicate greater anomaly diversity
> in recent IAS evaluations such as AnomalyAny, AnomalyDiffusion, and RealNet.
>
> This update also responds to Reviewer iR9j's request for a clearer mapping
> between evaluation metrics and IAS objectives. By separating reference-based
> LPIPS from IC-LPIPS, the revised manuscript clarifies that the same metric
> family can support different technical claims depending on whether the target is
> visual fidelity, editing consistency, background preservation, or anomaly
> diversity. We thank the reviewer again for this precise and helpful comment.

---

### Review · Reviewer_iR9j · 2026-06-26

**Summary Of Contributions:**

Summary:
This paper surveys Industrial Anomaly Synthesis (IAS) as a distinct research topic, rather than treating it only as an auxiliary component of industrial anomaly detection. It proposes a taxonomy that organizes existing IAS methods into four paradigms: hand-crafted synthesis, distribution hypothesis-based synthesis, generative model-based synthesis, and vision-language model-based synthesis. The paper further summarizes commonly used datasets, evaluation metrics, representative methods, practical trade-offs, and future research directions.

Strengths:
1. The topic is timely and relevant, as anomaly synthesis is becoming increasingly important for data augmentation, controllable inspection, and downstream anomaly detection in industrial scenarios.
2. The proposed taxonomy provides a clear and useful structure for organizing a diverse body of IAS methods, including both classical rule-based approaches and recent VLM-based methods.
3. The survey includes helpful summaries of datasets, evaluation metrics, practical characteristics, and future directions, making it a useful reference for researchers entering this area.

Weaknesses:
1. The survey methodology is not fully specified. The paper would be stronger if it described the literature search process, inclusion criteria, and coverage more explicitly.
2. Some boundaries between categories are not sufficiently justified, especially between GM-based and VLM-based synthesis, where diffusion models and prompt-based control may overlap.
3. The analysis is somewhat descriptive in places. More critical comparison of representative methods, including their assumptions, limitations, and suitable application scenarios, would improve the survey.
4. The evaluation discussion lists many metrics, but it could better explain which metrics are most appropriate for different IAS objectives, such as visual realism, detector improvement, or mask alignment.
5. Some tables and figures are dense and would benefit from clearer formatting, more self-contained captions, and more explicit explanations of symbols and abbreviations.

**Additional Comments:**

Overall, this is a timely and useful survey on an emerging topic. I am positive about the paper, provided that the authors improve the transparency of the survey methodology and sharpen several parts of the comparative analysis. The requested revisions are mostly clarifications and presentation improvements rather than fundamental changes.

**Audience:**

Yes

**Audience Explanation:**

The paper is likely to be of interest to researchers working on anomaly detection, industrial inspection, data augmentation, generative modeling, and controllable image synthesis. IAS is increasingly relevant for constructing training data, stress-testing anomaly detectors, and supporting segmentation-oriented industrial inspection. The paper’s taxonomy and summary of recent methods, especially VLM-based synthesis, should be useful to readers seeking a structured overview of this emerging topic.

**Broader Impact Concerns:**

I do not see major ethical concerns requiring a separate broader impact discussion.

**Claims And Evidence:**

Yes

**Claims Explanation:**

The main claims are generally supported by relevant literature, structured tables, and a clear organization of representative IAS methods. The proposed taxonomy is reasonable and helps clarify the development of the field from hand-crafted synthesis to generative and VLM-based approaches. The summaries of datasets, metrics, and practical trade-offs also provide useful evidence for the paper’s main narrative.

**Requested Changes:**

1. Please clarify the survey methodology, including the literature search process, inclusion/exclusion criteria, and the approximate scope of covered papers.
2. Please better justify the taxonomy boundaries, especially for methods that combine diffusion models, prompt conditioning, multimodal control, and local editing.
3. Please add more critical comparison across representative methods, focusing on their assumptions, practical limitations, and suitable application scenarios.
4. Please improve the readability of dense tables and figures by clarifying symbols, abbreviations, and captions, so that they are self-contained.

---

> ### Author Response · Authors · 2026-07-09
> **Response to Reviewer iR9j (Part 1/2)**
>
> We thank the reviewer for the careful and constructive assessment. We are encouraged that the reviewer finds the topic timely and the taxonomy useful. We agree that the original manuscript should more clearly describe the survey methodology, sharpen taxonomy boundaries, and provide more diagnostic comparisons. We have revised the manuscript accordingly.
>
> **Survey methodology, inclusion criteria, and coverage.**
> In Section 1, we added a paragraph describing the literature corpus and survey methodology. The revised manuscript explains that the corpus was constructed through keyword-based search, venue-oriented screening, and citation snowballing. We also list representative search keywords, including "industrial anomaly synthesis", "defect generation", "anomaly generation", "synthetic anomaly", "controllable anomaly synthesis", and "industrial defect image generation".
>
> We further specify the temporal scope, venue coverage, and inclusion/exclusion criteria. The revised survey mainly focuses on works available from 2019 to March 30, 2026 across major computer vision, machine learning, artificial intelligence, multimedia, and industrial inspection venues, while retaining earlier foundational studies and benchmarks when needed. We primarily include studies that introduce an explicit anomaly synthesis mechanism or synthesis-oriented component in industrial or closely related visual anomaly settings. These studies may synthesize abnormal images, image-mask pairs, anomaly masks, or anomaly-oriented latent/feature representations, either as standalone methods or as components of detection, localization, inspection, or segmentation frameworks. We exclude general anomaly detection studies with no clear relevance to industrial or related visual anomaly settings, and avoid counting multiple versions of the same work; when both a preprint and a peer-reviewed version are available, we retain the peer-reviewed version.
>
> To make the coverage more transparent, Table 1 now reports approximately 64 representative IAS methods, and Figure 2 shows the publication-source distribution. We also state that, because IAS is rapidly evolving and several 2026 works are concurrent, the survey is representative rather than exhaustive. These revisions address the reviewer’s request for a clearer search process, explicit inclusion/exclusion criteria, and a more transparent scope. This also relates to Reviewer AUfM’s request that recent and concurrent IAS works be acknowledged without implying exhaustive coverage through all of 2026.
>
> **Taxonomy boundaries, especially GM-based versus VLM-based synthesis.**
> We agree that the GM/VLM boundary was not sufficiently explicit. This issue overlaps with Reviewer AUfM’s comment on the GM/VLM boundary. We therefore revised Section 2.1, Table 2, and the opening of Section 6 to make the distinction clearer.
>
> In Section 2.1, we clarify that the taxonomy is not based on a single backbone or synthesis technique. Instead, the four paradigms are distinguished by dominant synthesis mechanism, knowledge source, dependence on large-scale pre-training, task adaptation strategy, and modality. The new Table 2 compares hand-crafted, distribution hypothesis-based, GM-based, and VLM-based synthesis along these criteria.
>
> The revised text states that GM-based methods may use GANs, VAEs, or diffusion models, but their synthesis capability mainly comes from task-specific training or adaptation on industrial data. In contrast, VLM-based methods build on large-scale pre-trained vision-language, multimodal, or text-conditioned generative priors, and adapt them to IAS through prompting, instruction tuning, lightweight adaptation, LoRA, textual inversion, few-shot reference adaptation, or other multimodal and spatial cues. Thus, the GM/VLM boundary lies mainly in pre-training dependence, adaptation paradigm, knowledge-transfer strategy, and modality, rather than in the use of diffusion models or the ability to synthesize abnormal images.
>
> We also revised the opening of Section 6 to restate this distinction. The manuscript now clarifies that architectural overlap does not by itself determine the taxonomy: GM-based and VLM-based methods may both adopt diffusion backbones, perform local editing, or generate image-mask pairs. Instead, the distinction follows the dominant synthesis paradigm described in Section 2.1, especially the primary source of synthesis capability, dependence on large-scale pre-training, adaptation strategy, knowledge-transfer mechanism, and modality. Methods are categorized as VLM-based when large-scale pre-trained vision-language, multimodal, or text-conditioned priors play a central role in transferring semantic or multimodal knowledge to IAS, rather than merely serving as an interchangeable generative backbone.

---

> ### Author Response · Authors · 2026-07-09
> **Response to Reviewer iR9j (Part 2/2)**
>
> **More critical comparison of assumptions, limitations, and suitable scenarios.**
> We agree that some parts of the original analysis were more descriptive than diagnostic. This concern overlaps with Reviewer AUfM’s request for failure-mode analysis and deployment constraints. To address both comments, we revised Sections 3--6 and Tables 5--8 so that each paradigm now discusses not only how anomalies are synthesized, but also when they are suitable and when they may fail.
>
> (1) Hand-crafted synthesis: Section 3 now explains that self-contained synthesis is suitable for rapid lightweight augmentation, but has limited diversity and may introduce synthetic artifacts. External-dependent synthesis supports broad appearance diversification, but can suffer from boundary inconsistency and semantic mismatch. Inpainting-based synthesis fits reconstruction-oriented pipelines, but may bias anomalies toward missing or corrupted appearances. We also added CAI as a recent cross-domain anomaly injection example.
>
> (2) Distribution hypothesis-based synthesis: Section 4 now distinguishes prior-dependent and data-driven methods more diagnostically. Prior-dependent methods are suitable when normal representations have stable geometry, but may fail when this assumption is unstable. Data-driven methods are useful for feature-level perturbation, but less suitable when deployment requires photorealistic image-space anomaly generation or reusable synthetic datasets.
>
> (3) GM-based synthesis: Section 5 clarifies the deployment objective of each subcategory. Full-image synthesis supports whole-image dataset expansion, but may suffer from mode collapse and domain overfitting. Full-image translation supports normal-to-abnormal translation, but has weak spatial controllability and is limited by learned domains. Local anomaly synthesis supports segmentation-oriented and spatially controlled augmentation, but may introduce boundary artifacts and depends on mask quality.
>
> (4) VLM-based synthesis: Section 6 explains that single-stage VLM-based methods are suitable for rapid multimodal-conditioned synthesis, but can suffer from semantic misalignment and weak masks. Multi-stage methods support workflows requiring synthesized masks, aligned image-mask pairs, anomaly-part synthesis, or progressive refinement, but may suffer from error propagation and pipeline complexity.
>
> Tables 5--8 now include "Suitable scenario" and "Practical limitation" columns, and Section 7 further connects these profiles to cross-paradigm deployment guidance. The revised manuscript therefore presents the paradigms not only as a method inventory, but also as a diagnostic comparison of assumptions, failure modes, and application boundaries.
>
> **Evaluation metrics and IAS objectives.**
> In Table 4 and Section 2.2, we revised the metric discussion to better connect evaluation metrics with IAS objectives. The revised discussion organizes evaluation into three perspectives: direct synthesis quality, downstream detection utility, and spatial localization accuracy. Direct synthesis metrics are tied to realism, fidelity, diversity, background preservation, or mask-conditioned control. Downstream detection metrics are tied to whether synthetic data improves detector training. Localization and segmentation metrics are used for spatial correspondence or pixel-level supervision.
>
> In response to Reviewer 78DH’s related comment on LPIPS, we revised Table 4 and Section 2.2. The previous single "LPIPS ↓" entry has been split into reference-based LPIPS, where lower values indicate better perceptual fidelity or editing consistency, and IC-LPIPS, where higher values indicate greater anomaly diversity. Thus, the LPIPS revision directly addresses Reviewer 78DH’s metric-specific concern and also supports Reviewer iR9j’s request that metrics be explained according to different IAS objectives.
>
> **Readability of tables, figures, symbols, and captions.**
> We improved the self-contained presentation of tables and figures. Table 2 defines the paradigm-level comparison criteria. Tables 5--8 now include notes clarifying the qualitative nature of the ratings, Train/Infer/Memory, and Control/Realism; Table 5 additionally defines N/A for training-free hand-crafted methods. Section 7 has been retitled "Practical Guidance and Future Directions," and Tables 9--10 provide a clearer view of input requirements, output forms, cost, controllability, downstream tasks, and key trade-offs. In Table 9, we replaced compact symbols with "Required", "Optional", and "--", and added notes explaining spatial annotation, reference input, and "(+ mask)". In Table 10, we added a note that cost excludes upstream foundation-model pre-training and that C/D/S denote classification, detection, and segmentation. These revisions also support Reviewer AUfM’s request to connect the taxonomy with practical deployment scenarios.
>
> We appreciate the reviewer’s suggestions and we would be happy to further clarify these revisions if needed.

---

### Author Response · Authors · 2026-07-09
**Summary of Revisions**

We thank the reviewers for their careful reading and constructive comments, which helped us improve the clarity, scope, and practical value of the survey.

Across the revision, we strengthened the manuscript in six connected ways:

(1) clarifying the literature corpus and survey methodology by adding the search strategy, temporal scope, venue coverage, inclusion/exclusion criteria, and representative nature of the reviewed IAS methods;

(2) sharpening the proposed taxonomy, especially the boundary between GM-based and VLM-based synthesis, by emphasizing differences in knowledge source, dependence on large-scale pre-training, task adaptation strategy, knowledge-transfer mechanism, and modality;

(3) making the paradigm-level discussions more diagnostic by expanding the discussion of suitable application scenarios, deployment boundary conditions, and practical failure modes of different IAS subcategories;

(4) revising the evaluation-metric discussion, especially the interpretation of LPIPS, to better reflect IAS-specific evaluation protocols and to distinguish perceptual fidelity from anomaly diversity;

(5) improving the readability and self-containedness of the tables by clarifying symbols, qualitative ratings, input requirements, output forms, practical limitations, and cross-paradigm comparison criteria; and

(6) updating recent and concurrent IAS-related works, including the published DualAnoDiff citation and several AAAI 2026 IAS-related methods, while clarifying that the survey is intended to be representative rather than exhaustive.

We have uploaded a revised manuscript PDF incorporating these changes. We hope the revised manuscript and the detailed responses below adequately address the reviewers’ concerns, and we sincerely thank the reviewers again for their helpful feedback.